# Structures and Bioactivities of Quadrangularisosides A, A_1_, B, B_1_, B_2_, C, C_1_, D, D_1_–D_4_, and E from the Sea Cucumber *Colochirus quadrangularis*: The First Discovery of the Glycosides, Sulfated by C-4 of the Terminal 3-*O*-Methylglucose Residue. Synergetic Effect on Colony Formation of Tumor HT-29 Cells of these Glycosides with Radioactive Irradiation

**DOI:** 10.3390/md18080394

**Published:** 2020-07-28

**Authors:** Alexandra S. Silchenko, Anatoly I. Kalinovsky, Sergey A. Avilov, Pelageya V. Andrijaschenko, Roman S. Popov, Pavel S. Dmitrenok, Ekaterina A. Chingizova, Svetlana P. Ermakova, Olesya S. Malyarenko, Salim Sh. Dautov, Vladimir I. Kalinin

**Affiliations:** 1G.B. Elyakov Pacific Institute of Bioorganic Chemistry, Far Eastern Branch of the Russian Academy of Sciences, Pr. 100-letya Vladivostoka 159, Vladivostok 690022, Russia; kaaniv@pidoc.dvo.ru (A.I.K.); avilov-1957@mail.ru (S.A.A.); pandrijashchenko@mail.ru (P.V.A.); prs_90@mail.ru (R.S.P.); paveldmt@piboc.dvo.ru (P.S.D.); martyyas@mail.ru (E.A.C.); swetlana_e@mail.ru (S.P.E.); vishchuk@mail.ru (O.S.M.); kalininv@piboc.dvo.ru (V.I.K.); 2A.V. Zhirmunsky National Scientific Center of Marine Biology, Far Eastern Branch, Russian Academy of Sciences, 17 Palchevskogo Street, Vladivostok 690041, Russia; daut49shakir@mail.ru

**Keywords:** *Colochirus quadrangularis*, triterpene glycosides, quadrangularisosides, sea cucumber, cytotoxic activity

## Abstract

Thirteen new mono-, di-, and trisulfated triterpene glycosides, quadrangularisosides A–D_4_ (**1−13**) have been isolated from the sea cucumber *Colochirus quadrangularis,* which was collected in Vietnamese waters. The structures of these glycosides were established by 2D NMR spectroscopy and HR-ESI (High Resolution Electrospray Ionization) mass spectrometry. The novel carbohydrate moieties of quadrangularisosides D–D_4_ (**8**−**12**), belonging to the group D, and quadrangularisoside E (**13**) contain three sulfate groups, with one of them occupying an unusual position—at C(4) of terminal 3-*O*-methylglucose residue. Quadrangularisosides A (**1**) and D_3_ (**11**) as well as quadrangularisosides A_1_ (**2**) and D_4_ (**12**) are characterized by the new aglycones having 25-hydroperoxyl or 24-hydroperoxyl groups in their side chains, respectively. The cytotoxic activities of compounds **1**–**13** against mouse neuroblastoma Neuro 2a, normal epithelial JB-6 cells, erythrocytes, and human colorectal adenocarcinoma HT-29 cells were studied. All the compounds were rather strong hemolytics. The structural features that most affect the bioactivity of the glycosides are the presence of hydroperoxy groups in the side chains and the quantity of sulfate groups. The membranolytic activity of monosulfated quadrangularisosides of group A (**1**, **2**) against Neuro 2a, JB-6 cells, and erythrocytes was relatively weak due to the availability of the hydroperoxyl group, whereas trisulfated quadrangularisosides D_3_ (**11**) and D_4_ (**12**) with the same aglycones as **1**, **2** were the least active compounds in the series due to the combination of these two structural peculiarities. The erythrocytes were more sensitive to the action of the glycosides than Neuro 2a or JB-6 cells, but the structure–activity relationships observed for glycosides **1**–**13** were similar in the three cell lines investigated. The compounds **3**−**5**, **8**, and **9** effectively suppressed the cell viability of HT-29 cells. Quadrangularisosides A_1_ (**2**), C (**6**), C_1_ (**7**), and E (**13**) possessed strong inhibitory activity on colony formation in HT-29 cells. Due to the synergic effects of these glycosides (0.02 μM) and radioactive irradiation (1 Gy), a decreasing of number of colonies was detected. Glycosides **1**, **3,** and **9** enhanced the effect of radiation by about 30%.

## 1. Introduction

Triterpene glycosides, biosynthesized by the sea cucumbers, are well-known secondary metabolites that are characterized by tremendous structural diversity [1,2,3,4], different kinds of biological activities [5,6], and taxonomic specificity [7,8,9,10]. Hence, the studies on this class of metabolites are still relevant.

The chemical investigation of the sea cucumber *Colochirus quadrangularis* was started earlier and the structures of some aglycones and glycosides were established [11,12,13,14,15]. 

Hence, it seemed very interesting to reinvestigate the composition of triterpene glycosides in the sea cucumber *Colochirus quadrangularis* to compare structural data reported earlier with those obtained by us. The animals were collected near the Vietnamese sea shore. Herein, we report the isolation and structure elucidation of 13 new glycosides having holostane-type aglycones and tetrasaccharide carbohydrate chains with one, two, and three sulfate groups. The structures of compounds named quadrangularisosides A (**1**), A_1_ (**2**), B (**3**), B_1_ (**4**), B_2_ (**5**), C (**6**), C_1_ (**7**), D (**8**), D_1_–D_4_ (**9**–**12**), and E (**13**) were established by the analyses of the ^1^H, ^13^C NMR, 1D TOCSY (Total Correlation Spectroscopy), and 2D NMR (^1^H,^1^H-COSY, HMBC, HSQC, ROESY—^1^H,^1^H-Correlated Spectroscopy, Heteronuclear Multiple Bond Correlation, Heteronuclear Single Quantum Correlation, Rotating-Frame Overhauser Effect Spectroscopy) spectra as well as HR-ESI ((High Resolution Electrospray Ionization)) mass spectra. The original spectra are presented in Appendix A. The hemolytic activities against mouse erythrocytes, cytotoxic activities against mouse neuroblastoma Neuro 2a, and normal epithelial JB-6 cells as well as the influence of **1**–**13** on the cell viability and colony formation of HT-29 cells have been studied.

## 2. Results and Discussion

### 2.1. Structures of the Earlier Published Glycosides from C. quadrangularis 

In 2004, the group of Chinese researches reported the structure elucidation of the aglycones obtained by an acid hydrolysis from the sum of glycosides of this species of sea cucumbers [11]. However, the paper contained some inaccuracies. Firstly, the specific name of the sea cucumber was written erroneously (*quadrangulaSis* instead of *quadrangulaRis*). Moreover, the assignment to genus *Pentacta* was also incorrect, since the actual specific name of this sea cucumber indexed by the “WoRMS” database is *Colochirus quadrangularis*. Secondly, thorough analysis of the NMR data provided in this paper for philinopgenin B possessing, according to the authors, 18(16)-lactone and unique 20(25)-epoxy-fragment, has raised many questions. If an 18(16)-lactone ring was present in the genin, the signals of H-16 and H-17 would be observed as the broad singlets due to the very small value of coupling constant *J*_17/16_ [16,17]. However, in the ^1^H NMR spectrum of philinopgenin B, the signal of H-16 was observed as a triplet, but its *J* value was not provided, and the signal of H-17 was observed as a multiplet. Moreover, the signals of carbons adjacent to the oxygen in the epoxy fragment should be deshielded to δ_C_ approximately 80–85 [17,18] whenever in the paper [11] the signals assigned to C-20 and C-25 were observed at δ_C_ 73.0 and 71.8, correspondingly. Finally, it is inadmissible to provide only integer values of *m*/*z* without any decimals as HR-ESI-MS data. Considering all these errors, the structure of philinopgenin B looks doubtful.

More recently, the structures of four glycosides, philinopsides A, B [12], E, and F [13], isolated from *Pentacta* (=*Colochirus*) *quadrangularis* were discussed by the same researchers. Philinopside B was described as a disulfated tetraoside with an uncommon position of the second sulfate group at C-2 of the xylose (the third residue in the carbohydrate chain) [12]. However, the comparison of the ^13^C NMR spectra of this glycoside and its desulfated derivative showed the absence of any differences in the δ_C_ values assigned to the xylose, occupying the third position in the carbohydrate chain, and therefore, the presence of the only sulfate group in the sugar chain of philinopside B—at C-4 of the first (xylose) residue. Moreover, the HR mass-spectrometric data have not been provided in the paper. The *m*/*z* of fragment ions observed in the ESI MS did not prove the presence of two sulfate groups in the compound.

Afterwards, the structures of pentactasides B and C were established as disulfated tetraosides with one of sulfate groups attached to C-2 of the quinovose residue [14], pentactasides I, II—as triosides lacking a 3-*O*-methylglucose unit, thus having terminal xylose residue [15] and, finally, pentactaside III—as a bioside with the carbohydrate chain identical to that of holothurins of group B [15,19]. The trisaccharide chains of pentactasides I and II are first found in the sea cucumber glycosides. The authors stated that additional studies are required for the final proving of their structures [15]. Actually, the analysis of the ^13^C NMR spectrum of pentactaside I showed that the signal of C-3 of the terminal (xylose) residue was deshielded (δ_C_ 87.3) and the signals of C-2 and C-4 of the same residue were shielded (δ_C_ 73.7 and 69.2, correspondingly) as compared with the spectrum of pentactaside II, which is characteristic for the glycosylation effects. So, doubts arise concerning the correctness of the structure of the glycoside.

### 2.2. Structural Elucidation of the Glycosides

The concentrated ethanolic extract of the sea cucumber *Colochirus quadrangularis* was chromatographed on a Polychrom-1 column (powdered Teflon, Biolar, Latvia). The glycosides were eluted with 50% EtOH and separated by chromatography on a Si gel column using CHCl_3_/EtOH/H_2_O (100:100:17) and (100:125:25) as mobile phases. The obtained fractions were subsequently subjected to HPLC on a silica-based Supelcosil LC-Si (4.6 × 150 mm) column and on a reversed-phase semipreparative Supelco Discovery HS F5-5 (10 × 250 mm) column to yield compounds **1**–**13** (Figure 1) along with nine known earlier glycosides: colochirosides B_1_, B_2_, and B_3_ [20], lefevreosides A_2_ and C [21], neothyonidioside [22], hemoiedemoside A [23], and philinopsides A [12] and F [13], which were isolated earlier from different species of sea cucumbers. The known compounds were identified by the comparison of their ^1^H and ^13^C NMR spectra with those reported in the literature.

The ^1^H and ^13^C NMR spectra corresponding to the carbohydrate chains of quadrangularisosides A (**1**) and A_1_ (**2**) (Table 1) were identical to each other and to those of known compounds isolated from this species: colochirosides B_1_, B_2_, and B_3_, lefevreosides A_2_ and C, neothyonidioside and philinopside A, having linear tetrasaccharide monosulfated carbohydrate moieties with the xylose residue as the third unit. Such a sugar chain is common in the glycosides of sea cucumbers of different taxa [2].

The molecular formula of quadrangularisoside A (**1**) was determined to be C_55_H_85_O_27_SNa from the [M_Na_ − Na]**^−^** ion peak at *m*/*z* 1209.5004 (calc. 1209.5004) in the (−)HR-ESI-MS and [M_Na_ + Na]^+^ ion peak at *m*/*z* 1255.4779 (calc. 1255.4789) in the (+)HR-ESI-MS. The analysis of the ^13^C and ^1^H NMR spectra of the aglycone part of **1** suggested the presence of an 18(20)-lactone [from the signals of C(18) at δ_C_ 180.0 and C(20) at δ_C_ 84.8], a 7(8)-double bond [from the signal of secondary carbon C(7) at δ_C_ 120.3 and the corresponding proton signal at δ_H_ 5.61 (m, H(7)), the signal of quaternary carbon C(8) at δ_C_ 145.5] as well as the *β*-*O*Ac group at C(16) [from the signal of carbon at δ_C_ 74.8 (C(16)) and the corresponding proton signal at δ_H_ 5.92 (H(16), q, *J* = 8.5 Hz) along with the signals of two carbons of the *O*-acetic group at δ_C_ 170.7 and 21.1 and a signal of protons of the methyl group at δ_H_ 1.97(s)] (Table 2). All these data indicated the presence of a holostane-type nucleus in **1**. The signals of olefinic carbons at δ_C_ 124.2 (C(23)) and 139.5 (C(24)) indicated the presence of a double bond in the side chain of quadrangularisoside A (**1**). The δ_C_ values of the olefinic carbons were close to those in the ^13^C NMR spectrum of psolusoside D_3_ [24] that allowed supposing the 23(24)-position of the double bond. The HMBC correlations from methyl groups H(26) and H(27) to the olefinic carbons C(23) and C(24) corroborated the position of the double bond. The coupling pattern of olefinic protons H(23) (δ_H_ 5.71, dt, *J* = 6.6; 15.5 Hz) and H(24) (δ_H_ 5.97, d, *J* = 15.8 Hz) indicated a 23*E*-configuration of the double bond in **1**. The signal of the tertiary bearing oxygen carbon C(25) was deduced from the HMBC correlations between the signals of methyl groups H(26) (δ_H_ 1.46, s) and H(27) (δ_H_ 1.48, s) and the signal at δ_C_ 81.3. This chemical shift was also close to that of the corresponding carbon (δ_C_ 80.8, C(25)) in the spectrum of psolusoside D_3_ [24]. So, the NMR and HR-ESI-MS data indicated the presence of a 25-hydroperoxy-23*E*-ene fragment in the side chain of **1**. This is the third finding of a hydroperoxyl group in the triterpene glycosides aglycones of sea cucumbers [24,25].

The (+)ESI-MS/MS of **1** demonstrated the fragmentation of the [M_Na_ + Na]^+^ ion at *m*/*z* 1255.5. The peaks of fragment ions were observed at *m*/*z* 1223.5 [M_Na_ + Na − OOH + H]^+^, 1103.5 [M_Na_ + Na − OOH − NaSO_4_ + H]^+^, 927.5 [M_Na_ + Na − OOH − NaSO_4_ − C_7_H_12_O_5_ (MeGlc) + H]^+^, 795.4 [M_Na_ + Na − OOH − NaSO_4_ − C_7_H_12_O_5_ (MeGlc) − C_5_H_8_O_4_ (Xyl) + H]^+^, 729.2 [M_Na_ + Na − C_32_H_47_O_6_ (Agl) + H]^+^, 649.3 [M_Na_ + Na− OOH − NaSO_4_ − C_7_H_12_O_5_ (MeGlc) − C_5_H_8_O_4_ (Xyl) − C_6_H_10_O_4_ (Qui) + H]^+^, 609.2 [M_Na_ + Na − C_32_H_47_O_6_ (Agl) − NaHSO_4_]^+^, 477.1 [M_Na_ + Na − C_32_H_47_O_6_ (Agl) − NaHSO_4_ − C_5_H_8_O_4_ (Xyl) + H]^+^, corroborating the structure of quadrangularisoside A (**1**).

All these data indicate that quadrangularisoside A (**1**) is 3*β*-*O*-[3-*O*-methyl-*β*-d-glucopyranosyl-(1→3)-*β*-d-xylopyranosyl-(1→4)-*β*-d-quinovopyranosyl-(1→2)-4-*O*-sodium sulfate-*β*-d-xylopyranosyl]-25-peroxy-16*β*-acetoxyholosta-7,23*E*-diene.

The molecular formula of quadrangularisoside A_1_ (**2**) was determined to be the same as that of **1** (C_55_H_85_O_27_SNa) from the [M_Na_ − Na]**^−^** ion peak at *m*/*z* 1209.5006 (calc. 1209.5004) in the (−)HR-ESI-MS and [M_Na_ + Na]**^+^** ion peak at *m*/*z* 1255.4772 (calc. 1255.4789) in the (+)HR-ESI-MS. The signals in the ^13^C NMR spectrum of **2** assigning to triterpene nucleus (C(1)–C(20), C(30)–C(32)) coincided with the corresponding signals in the spectrum of **1** indicating the difference of these glycoside only in the side chains structures (Table 3). An isolated spin system formed by the protons H(22)–H(24) was deduced from the ^1^H,^1^H-COSY spectrum of **2**. The signal of H(24) was deshielded and observed at δ_H_ 4.51 (brt, *J* = 6.2 Hz) due to the attachment of a hydroperoxyl group to C(24) in the glycoside **2**, which was corroborated by the deshielding of the signal of C(24) to δ_C_ 89.2. The signals of olefinic carbons at δ_C_ 144.9 (C(25)) and 113.6 (C(26) as well as olefinic protons at δ_H_ 5.14 (brs, H(26)) and 5.02 (brs, H(26′)) indicated the presence of a terminal double bond in the side chain of **2**. The positions of a double bond and hydroperoxyl group were confirmed by the HMBC correlations H(24)/C(26); H(26) and H(26′)/C(24), C(27); H(27)/C(24), C(25), C(26). Hence, quadrangularisosides A (**1**) and A_1_ (**2**) are the isomers by the position of the hydroperoxy-ene fragment in their side chains.

The 24(*S*)-configuration was assigned to the 24(*S*)-hydroxy-25-dehydroechinoside A isolated earlier from the sea cucumber *Actinopyga flammea* [26]. The same configuration of C(24)-stereocenter was established by Mosher′s method in the aglycone of cucumarioside A_7_ from the sea cucumber *Eupentacta fraudatrix* [27], which differs from the aglycone of **2** only by a hydroxyl substituent at C(24) instead of a hydroperoxyl group. Thus, 24(*S*)-configuration can be attributed to the aglycone of quadrangularisoside A_1_ (**2**) based on its biogenetic background.

The (+) ESI-MS/MS of **2** demonstrated the fragmentation of the [M_Na_ + Na − OOH]^+^ ion at *m*/*z* 1237.5 corresponding to a dehydrated molecule, whose formation was caused by the presence of a hydroperoxyl group in the aglycone. The peaks of fragment ions were observed at *m*/*z* 1177.4 [M_Na_ + Na − OOH − CH_3_COOH]^+^, 1117.5 [M_Na_ + Na − OOH − NaHSO_4_]^+^. The fragmentation of the [M_Na_ + Na]^+^ ion at *m*/*z* 1237.5 resulted in the formation of fragment ions at the same *m*/*z* as in **1**: 729.2 [M_Na_ + Na− C_32_H_47_O_6_ (Agl) + H]^+^, 609.2 [M_Na_ + Na − C_32_H_47_O_6_ (Agl) − NaHSO_4_]^+^, 477.1 [M_Na_ + Na − C_32_H_47_O_6_ (Agl) − NaHSO_4_ − C_5_H_8_O_4_ (Xyl) + H]^+^, corroborating their isomerism.

All these data indicate that quadrangularisosides A_1_ (**2**) is 3*β*-*O*-[3-*O*-methyl-*β*-d-glucopyranosyl-(1→3)-*β*-d-xylopyranosyl-(1→4)-*β*-d-quinovopyranosyl-(1→2)-4-*O*-sodium sulfate-*β*-d-xylopyranosyl]-24*S*-peroxy-16*β*-acetoxyholosta-7,25-diene.

The ^1^H and ^13^C NMR spectra corresponding to the carbohydrate chains of quadrangularisosides B (**3**), B_1_ (**4**), and B_2_ (**5**) (Table 4) were identical to each other and coincided with the spectra of the carbohydrate part of pseudostichoposide B isolated earlier from the sea cucumber *Pseudostichopus trachus* (=*P. mollis*) (family Pseudostichopodidae, order Persiculida) [28] and violaceusoside E from *Pseudocolochirus violaceus* [29]. This linear disulfated tetrasaccharide chain was characterized by the attachment of the second sulfate group to C(3) of the quinovose residue. The structure of the carbohydrate chain of glycosides **3**–**5** was established by the thorough analysis of the ^1^H,^1^H-COSY, HSQC, and 1D TOCSY spectra for each monosaccharide unit, and the positions of interglycosidic linkages were elucidated based on the ROESY and HMBC correlations (Table 4). The finding of sugar moieties of the glycosides with the sulfate group at C(3) of the quinovose is very rare. Only three glycosides having such structural features are found so far [28,29] among more than 150 known triterpene glycosides from the sea cucumbers.

It is necessary to note that pentactasides B and C were reported earlier to have the disulfated tetrasaccharide chains with the same monosacaccharide composition as in **3**–**5** but differing by the position of a second sulfate group that was positioned at C(2) of the quinovose residue [14]. The analysis of the NMR data provided in the paper [14] allowed supposing the erroneous interpretation of these data by the authors. When the sulfate group is bonded to C(2) of the monosaccharide residue in the pyranose form, it resulted in the up-field shifting of the signal of anomeric carbon to δ_C_ approximately 100 [30]. Whenever the signals of C(1) of a quinovose unit in pentactasides B and C were observed, δ_C_ 102.5 and 102.9, correspondingly, indicating the absence of a sulfate group at C(2) of this residue. So, the signals of C(2) at δ_C_ 81.1 and 81.4 and C(3) at δ_C_ 74.5 and 75.0 of the quinovose units in pentactasides B and C, correspondingly, probably were displaced with each other, and the second sulfate group is located at C(3)Qui2 in these glycosides. Therefore, the structures of pentactasides B and C most likely are identical to those of quadrangularisosides B (**3**) and B_1_ (**4**), correspondingly.

The molecular formula of quadrangularisoside B (**3**) was determined to be C_55_H_84_O_28_S_2_Na_2_ from the [M_2Na_ − Na]**^−^** ion peak at *m*/*z* 1279.4489 (calc. 1279.4494) and [M_2Na_ − 2Na]^2**−**^ ion peak at *m*/*z* 628.2311 (calc. 628.2301) in the (−)HR-ESI-MS as well as from the [M_2Na_ + Na]**^+^** ion peak at *m*/*z* 1325.4272 (calc. 1325.4278) in the (+)HR-ESI-MS. The signals in the ^13^C NMR spectrum of the aglycone part of **3** were very close to those in the spectrum of cucumarioside A_1_ from *E. fraudatrix* [28] and pentactaside B [14], indicating the identity of their aglycones (Table 5). This holostane-type aglycone with 7(8)- and 24(25)-double bonds and a 16*β*-acetoxy-group is quite common for the glycosides of sea cucumbers of the order Dendrochirotida [2,4,5].

The (−)ESI-MS/MS of **3** showed the fragmentation of the [M_2Na_ − Na]^−^ ion at *m*/*z* 1279.5. The peaks of fragment ions were observed at *m*/*z* 1219.4 [M_2Na_ − Na − CH_3_COOH]^−^, 1177.5 [M_2Na_ − Na − NaSO_3_ + H]^−^. The (+)ESI-MS/MS of **3** showed that the fragmentation of the [M_2Na_ + Na]^+^ ion at *m*/*z* 1325.4 resulted in the formation of fragment ions whose peaks were observed at *m*/*z* 1223.5 [M_2Na_ + Na − NaSO_3_ + H]^+^, 915.4 [M_2Na_ + Na − NaSO_3_ − C_7_H_12_O_5_ (MeGlc) − C_5_H_8_O_4_ (Xyl) + H]^+^, 813.4 [M_2Na_ + Na − C_32_H_47_O_5_ (Agl) − H]^+^, 711.1 [M_2Na_ + Na − C_32_H_47_O_5_ (Agl) − SO_3_Na]^+^, 579.1 [M_2Na_ + Na − C_32_H_47_O_5_ (Agl) − C_5_H_7_O_7_SNa (XylSO_3_Na) − H]^+^, 535.3 [M_2Na_ + Na − C_32_H_47_O_5_ (Agl) − SO_3_Na − C_7_H_12_O_5_ (MeGlc)]^+^, 403.0 [M_2Na_ + Na − C_32_H_47_O_5_ (Agl) − SO_3_Na − C_7_H_12_O_5_ (MeGlc) − C_5_H_8_O_4_ (Xyl)]^+^, and 331.1 [M_2Na_ + Na − C_32_H_47_O_5_ (Agl) − C_5_H_7_O_7_SNa (XylSO_3_Na) − C_6_H_9_O_7_SNa (QuiSO_3_Na) − H]^+^, which corroborate the monosaccharide sequence in the carbohydrate chain of **3**.

All these data indicate that quadrangularisoside B (**3**) is 3*β*-*O*-[3-*O*-methyl-*β*-d-glucopyranosyl-(1→3)-*β*-d-xylopyranosyl-(1→4)-3-*O*-sodium sulfate-*β*-d-quinovopyranosyl-(1→2)-4-*O*-sodium sulfate-*β*-d-xylopyranosyl]-16*β*-acetoxyholosta-7,24-diene.

The molecular formula of quadrangularisoside B_1_ (**4**) was determined to be C_55_H_84_O_28_S_2_Na_2_ from the [M_2Na_ − Na]**^−^** ion peak at *m*/*z* 1279.4502 (calc. 1279.4494) and [M_2Na_ − 2Na]^2**−**^ ion peak at *m*/*z* 628.2320 (calc. 628.2301) in the (−)HR-ESI-MS as well as from the [M_2Na_ + Na]**^+^** ion peak at *m*/*z* 1325.4272 (calc. 1325.4278) in the (+)HR-ESI-MS, which was coincident with the formula of quadrangularisoside B (**3**) and indicated their isomerism. In the ^1^H and ^13^C NMR spectra of the aglycone part of **4**, the signals of holostane-type aglycone having 7(8)- and 25(26)-double bonds as well as a 16*β*-acetoxy-group were observed (Table 6). This aglycone is identical to that of pentactasides C [14] and II [15] isolated earlier from the same species and is frequently occurred in the glycosides of the sea cucumbers of the order Dendrochirotida [2,4,5,21,30].

The (+)ESI-MS/MS of **4** showed the fragmentation of the [M_2Na_ + Na]^+^ ion at *m*/*z* 1325.4. The peaks of fragment ions were observed at *m*/*z* 1223.5 [M_2Na_ + Na − NaSO_3_ + H]^+^, 1205.5 [M_2Na_ + Na − NaHSO_4_]^+^, 1085.5 [M_2Na_ + Na − 2NaHSO_4_]^+^, 897.4 [M_2Na_ + Na − NaHSO_4_ − C_7_H_12_O_5_ (MeGlc) − C_5_H_8_O_4_ (Xyl)]^+^, 711.1 [M_2Na_ + Na − C_32_H_47_O_5_ (Agl) − SO_3_Na]^+^, 579.1 [M_2Na_ + Na − C_32_H_47_O_5_ (Agl) − C_5_H_7_O_7_SNa (XylSO_3_Na) − H]^+^, 403.0 [M_2Na_ + Na − C_32_H_47_O_5_ (Agl) − SO_3_Na − C_7_H_12_O_5_ (MeGlc) − C_5_H_8_O_4_ (Xyl)]^+^, and 331.1 [M_2Na_ + Na − C_32_H_47_O_5_ (Agl) − C_5_H_7_O_7_SNa (XylSO_3_Na) − C_6_H_9_O_7_SNa (QuiSO_3_Na) − H]^+^, corroborating the structure of quadrangularisoside B_1_ (**4**).

All these data indicate that quadrangularisoside B_1_ (**4**) is 3*β*-*O*-[3-*O*-methyl-*β*-d-glucopyranosyl-(1→3)-*β*-d-xylopyranosyl-(1→4)-3-*O*-sodium sulfate-*β*-d-quinovopyranosyl-(1→2)-4-*O*-sodium sulfate-*β*-d-xylopyranosyl]-16*β*-acetoxyholosta-7,25-diene.

The molecular formula of quadrangularisoside B_2_ (**5**) was determined to be C_53_H_80_O_27_S_2_Na_2_ from the [M_2Na_ − Na]**^−^** ion peak at *m*/*z* 1235.4243 (calc. 1235.4232) and [M_2Na_ − 2Na]^2**−**^ ion peak at *m*/*z* 606.2189 (calc. 606.2170) in the (−)HR-ESI-MS as well as the [M_2Na_ + Na]^+^ ion peak at *m*/*z* 1281.4004 (calc. 1281.4016) in the (+)HR-ESI-MS. The signals in the ^1^H and ^13^C NMR spectra of the aglycone part of **5** were characteristic of holostane-type aglycone having 9(11)- and 25(26)-double bonds and a 16-keto-group (Table 7). This aglycone is known as holotoxinogenin, and it was found first in the glycosides named as holotoxins A_1_ and B_1_, which were isolated from the sea cucumber *Apostichopus japonicus* (Stichopodidae, Synallactida) [31] and are rather common for the glycosides of the sea cucumbers of different orders [2,4,5].

The (+)ESI-MS/MS of **5** demonstrated the fragmentation of the [M_2Na_ + Na]^+^ ion at *m*/*z* 1281.4. The peaks of fragment ions were observed at *m*/*z* 1179.4 [M_2Na_ + Na − NaSO_3_ + H]^+^, 1059.5 [M_2Na_ + Na − NaSO_3_ − NaHSO_4_ + H]^+^, confirming the presence of two sulfate groups. The peaks of fragment ions at *m*/*z* 711.1, 579.1, 403.0, and 331.1 were the same as in the mass spectra of the glycosides **3** and **4** due to the identity of their carbohydrate chains.

All these data indicate that quadrangularisoside B_2_ (**5**) is 3*β*-*O*-[3-*O*-methyl-*β*-d-glucopyranosyl-(1→3)-*β*-d-xylopyranosyl-(1→4)-3-*O*-sodium sulfate-*β*-d-quinovopyranosyl-(1→2)-4-*O*-sodium sulfate-*β*-d-xylopyranosyl]-16-ketoholosta-7,25-diene.

The ^13^C NMR spectra of the carbohydrate moieties of quadrangularisosides C (**6**) and C_1_ (**7**) were identical to each other (Table 8) and to those of hemoiedemoside A isolated first from the sea cucumber *Hemoiedema spectabilis* [23] and identified by us in the glycosidic sum of *C. quadrangularis.* The glycosides **6** and **7** have a tetrasaccharide linear carbohydrate chain differing from that of **1** and **2** by the sulfated by C(6) glucose residue as the third unit in the chain instead of a xylose residue. The carbohydrate chain structure of quadrangularisosides C (**6**) and C_1_ (**7**) was elucidated based on a thorough analysis of 1D and 2D NMR spectra (Table 8). Besides compounds **6**, **7**, and hemoiedemoside A, such a sugar moiety is a part of another six glycosides [32,33,34,35,36,37] isolated from the sea cucumbers of the order Dendrochirotida.

The molecular formula of quadrangularisoside C (**6**) was determined to be C_56_H_86_O_29_S_2_Na_2_ from the [M_2Na_ − Na]**^−^** ion peak at *m*/*z* 1309.4608 (calc. 1309.4599) and [M_2Na_ − 2Na]^2**−**^ ion peak at *m*/*z* 643.2369 (calc. 643.2354) in the (−)HR-ESI-MS. In the (+)HR-ESI-MS, the [M_2Na_ + Na]^+^ ion peak observed at *m*/*z* 1355.4373 (calc. 1355.4384) corresponded to the same molecular formula. The signals of C(1)–C(4) and C(15)–C(32) in the ^13^C NMR spectrum of **6** (Table 9) were close to those in the spectrum of quadrangularisoside B_1_ (**4**), but the signals characteristic for the 9(11)-double bond in the triterpene nucleus [at δ_C_ 150.6 (C(9)) and 110.9 (C(11)) as well as δ_H_ 5.16 (m, H(11))] were observed instead of the signals assigned to the 7(8)-double bond in **4**. So, these compounds were the isomers by the double bond position in the aglycone nuclei. The aglycone identical to that of quadrangularisoside C (**6**) was earlier found only in cladoloside A_3_ from the sea cucumber *Cladolabes schmeltzii* [36].

The (−)ESI-MS/MS of **6** demonstrated the fragmentation of the [M_2Na_ − Na]^−^ ion at *m*/*z* 1309.5. The peaks of fragment ions were observed at *m*/*z* 1249.4 [M_2Na_ − Na − CH_3_COOH]^−^, 1189.5 [M_2Na_ − Na − NaHSO_4_]^−^, 1129.5 [M_2Na_ − Na − CH_3_COOH − NaHSO_4_]^−^, 677.2 [M_2Na_ − Na − NaHSO_4_ − C_32_H_47_O_5_ (Agl) − H]^−^, 563.1 [M_2Na_ − Na − NaHSO_4_ − C_32_H_47_O_5_ (Agl) − C_5_H_7_O_3_ (Xyl)]^−^, 417.1 [M_2Na_ − Na − NaHSO_4_ − C_32_H_47_O_5_ (Agl) − C_5_H_7_O_3_ (Xyl) − C_6_H_10_O_4_ (Qui)]^−^, and 241.0 [M_2Na_ − Na − NaHSO_4_ − C_32_H_47_O_5_ (Agl) − C_5_H_7_O_3_ (Xyl) − C_6_H_10_O_4_ (Qui) − C_7_H_12_O_5_ (MeGlc)]^−^.

All these data indicate that quadrangularisoside C (**6**) is 3*β*-*O*-[3-*O*-methyl-*β*-d-glucopyranosyl-(1→3)-6-O-sodium sulfate-*β*-d-glucopyranosyl-(1→4)-*β*-d-quinovopyranosyl-(1→2)-4-*O*-sodium sulfate-*β*-d-xylopyranosyl]-16*β*-acetoxyholosta-9(11),25-diene.

The molecular formula of quadrangularisoside C_1_ (**7**) was determined to be C_56_H_88_O_29_S_2_Na_2_ from the [M_2Na_ − Na] **^−^** ion peak at *m*/*z* 1311.4763 (calc. 1311.4756), [M_2Na_ − 2Na]^2**−**^ ion peak at *m*/*z* 644.2447 (calc. 644.2432), and [M_2Na_ + Na]**^+^** ion peak at *m*/*z* 1357.4520 (calc. 1357.4540) in the (−) and (+)HR-ESI-MS, correspondingly. The signals of carbons corresponding to the holostane-type nucleus in the ^13^C NMR spectrum of **7** (Table 10) were coincident with those in the spectra of quadrangularisosides A (**1**), A_1_ (**2**), B (**3**), and B_1_ (**4**), indicating the presence of the aglycone having a 7(8)-double bond and 16*β*-acetoxy-group. The signals of H(22)–H(27) form the isolated spin system in the ^1^H,^1^H-COSY spectrum of **7**. So, the presence of an unsubstituted saturated side chain was supposed for this compound. Actually, the ^13^C NMR spectrum of the aglycone part of **7** was coincident with that of lefevreoside A_2_, isolated first from *A. lefevrei* [21] and identified in the glycosidic sum of *C. quadrangularis*. The aglycone identical to that of quadrangularisoside C_1_ (**7**) is frequently occurred in the glycosides of representatives of the order Dendrochirotida [2,4,5].

The (−)ESI-MS/MS of **7** demonstrated the fragmentation of the [M_2Na_ − Na]^−^ ion at *m*/*z* 1311.5. The peaks of fragment ions were observed at *m*/*z* 797.1 [M_2Na_ − Na − C_32_H_49_O_5_ (Agl) − H]^−^, 677.2 [M_2Na_ − Na − NaHSO_4_ − C_32_H_49_O_5_ (Agl) − H]^−^, 563.1 [M_2Na_ − Na − NaHSO_4_ − C_32_H_49_O_5_ (Agl) − C_5_H_7_O_3_ (Xyl)]^−^, 417.1 [M_2Na_ − Na − NaHSO_4_ − C_32_H_49_O_5_ (Agl) − C_5_H_7_O_3_ (Xyl) − C_6_H_10_O_4_ (Qui)]^−^, and 241.0 [M_2Na_ − Na − NaHSO_4_ − C_32_H_49_O_5_ (Agl) − C_5_H_7_O_3_ (Xyl) − C_6_H_10_O_4_ (Qui) − C_7_H_12_O_5_ (MeGlc)]^−^, which were coincident with those in the MS/MS spectrum of **6**, confirming the identity of their carbohydrate chains.

All these data indicate that quadrangularisoside C_1_ (**7**) is 3*β*-*O*-[3-*O*-methyl-*β*-d-glucopyranosyl-(1→3)-6-O-sodium sulfate-*β*-d-glucopyranosyl-(1→4)-*β*-d-quinovopyranosyl-(1→2)-4-*O*-sodium sulfate-*β*-d-xylopyranosyl]-16*β*-acetoxyholost-7-ene.

The ^13^C NMR spectra of the carbohydrate moieties of quadrangularisosides D–D_4_ (**8**–**12**) were coincident to each other (Table 11), indicating the presence and identity of sugar moieties in these glycosides. In the ^1^H and ^13^C NMR spectra of the carbohydrate part of **8**–**12**, four characteristic doublets at δ_H_ 4.63–5.12 (*J* = 7.1–8.0 Hz), and corresponding to them, four signals of anomeric carbons at δ_C_ 102.8–104.6 were indicative of a tetrasaccharide chain and *β*-configurations of glycosidic bonds. The ^1^H,^1^H-COSY, HSQC, and 1D TOCSY spectra of **8**–**12** showed the signals of the isolated spin systems assigned to two xylose residues, one quinovose residue, and one 3-*O*-methylglucose residue. These data indicated the same monosaccharide composition of the sugar chain of **8**–**12** as in quadrangularisosides of the groups A (**1**, **2**) and B (**3**–**5**). The comparison of the ^13^C NMR spectra of **8**–**12** and **3**–**5** showed the coincidence of the signals corresponding to the monosaccharide residues from the first to the third. The signals, assigning to the terminal 3-*O*-methylglucose residue in **8**–**12**, were deduced by the analysis of ^1^H,^1^H-COSY and 1D TOCSY spectra. The signal of C(4) MeGlc4 was observed at δ_C_ 76.2, the signal of C(3) was observed at δ_C_ 85.2, and the signal of C(5) was observed at δ_C_ 76.3 in the spectra of **8**–**12**. Hence, the α- and β-shifting effects due to the attachment of a sulfate group to C(4) MeGlc4 of quadrangularisosides of the group D (**8**–**12**) were observed in comparison with the δ_C_ values of the corresponding signals in the spectra of quadrangularisosides of the group B (**3**–**5**) (δ_C_ 70.3 for C(4) MeGlc4, δ_C_ 86.8 for C(3) MeGlc4, and δ_C_ 77.4 for C(5) MeGlc4) having non-sulfated 3-*O*-methylglucose residue in the same position of the carbohydrate chain.

Hence, three sulfate groups are present in the carbohydrate chain of quadrangularisosides of the group D (**8**–**12**): at C(4) Xyl1—the common position of this functionality for the sea cucumber glycosides, at C(3) Qui2—the rare position of the group, and finally, at C(4) MeGlc4—the unique for the glycosides position of a sulfate group. The similar structural feature was found earlier in the carbohydrate chains of some glycosides from *Psolus fabricii*, but the sulfate group was attached to C(4) of the terminal disulfated glucose residue having an additional sulfate group at the C(2) or C(6) position [37] and in the carbohydrate chain of stichorrenoside B isolated from *Stichopus horrens* [38], where the sulfate group was attached to C(4) of the glucose residue terminal in its disaccharide chain. The majority of known sulfated glycosides contain terminal 3-*O*-methylglucose residue with the sulfate group attached to C(6); in contrast, the sulfation of C(4) of the 3-*O*-methylglucose residue in the linear tetrasaccharidechain as in compounds **8**–**12** has never been discovered earlier in the sea cucumbers glycosides.

The positions of interglycosidic linkages were established by the ROESY and HMBC spectra of **8**–**12** (Table 11) where the cross-peaks between H(1) of the xylose and H(3) (C(3)) of an aglycone, H(1) of the second residue (quinovose) and H(2) (C(2)) of the xylose, H(1) of the third residue (xylose) and H(4) (C(4)) of the second residue (quinovose), H(1) of the fourth residue (3-O-methylglucose) and H-3 (C(3)) of the third residue (xylose) were observed.

The molecular formula of quadrangularisoside D (**8**) was determined to be C_55_H_83_O_31_S_3_Na_3_ from the [M_3Na_ − Na]**^−^** ion peak at *m*/*z* 1381.3888 (calc. 1381.1881), [M_3Na_ − 2Na]^2−^ ion peak at *m*/*z* 679.2012 (calc. 679.1995), [M_3Na_ − 3Na]^3−^ ion peak at *m*/*z* 445.1387 (calc. 445.1366), and [M_3Na_ + Na]**^+^** ion peak at *m*/*z* 1427.3663 (calc. 1427.3666) in the (−) and (+) HR-ESI-MS, respectively. The ^13^C NMR spectrum of the aglycone part of **8** (Table 5) coincided with that of quadrangularisoside B (**3**), indicating the identity of their aglycones.

The (−)ESI-MS/MS of quadrangularisoside D (**8**) demonstrated the fragmentation of the [M_3Na_ − Na]^−^ ion at *m*/*z* 1381.4. The peaks of fragment ions were observed at *m*/*z* 1261.4 [M_3Na_ − Na − NaHSO_4_]^−^, 1103.4 [M_3Na_ − Na − C_7_H_11_O_8_SNa (MeGlcSO_3_Na)]^−^, 751.3 [M_3Na_ − Na −NaHSO_4_− C_32_H_47_O_5_ (Agl) + H]^−^, 533.1 [M_3Na_ − Na − 2NaSO_3_ − C_33_H_47_O_5_ (Agl) − C_5_H_8_O_4_ (Xyl) + H]^−^, 387.1 [M_3Na_ − Na − 2NaSO_3_ − C_33_H_47_O_5_ (Agl) − C_5_H_8_O_4_ (Xyl) − C_6_H_10_O_4_ (Qui) + H]^−^, and 255.0 [M_3Na_ − Na − 2NaSO_3_ − C_33_H_47_O_5_ (Agl) − C_5_H_8_O_4_ (Xyl) − C_6_H_10_O_4_ (Qui) − C_5_H_8_O_4_ (Xyl) + 2H]^−^.

All these data indicate that quadrangularisoside D (**8**) is 3*β*-*O*-[4-*O*-sodium sulfate-3-*O*-methyl-*β*-d-glucopyranosyl-(1→3)-*β*-d-xylopyranosyl-(1→4)-3-*O*-sodium sulfate-*β*-d-quinovopyranosyl-(1→2)-4-*O*-sodium sulfate-*β*-d-xylopyranosyl]-16*β*-acetoxyholosta-7,24-diene.

The molecular formula of quadrangularisoside D_1_ (**9**) was determined to be the same as for quadrangularisoside D (**8**) (C_55_H_83_O_31_S_3_Na_3_) from the [M_3Na_ − Na]**^−^** ion peak at *m*/*z* 1381.3891 (calc. 1381.3881), [M_3Na_ − 2Na]^2**−**^ ion peak at *m*/*z* 679.2015 (calc. 679.1995), and [M_3Na_ − 3Na]^3**−**^ ion peak at *m*/*z* 445.1390 (calc. 445.1366) in the (−)HR-ESI-MS as well as from the [M_3Na_ + Na]**^+^** ion peak at *m*/*z* 1427.3653 (calc. 1427.3666) in the (+)HR-ESI-MS. The aglycone of quadrangularisoside D_1_ (**9**) was identical to that of quadrangularisoside B_1_ (**4**), which was deduced from the coincidence of their ^13^C NMR spectra (Table 6).

The (−)ESI-MS/MS of **9** demonstrated the fragmentation of the [M_3Na_ − Na]^−^ ion at *m*/*z* 1381.4. The peaks of fragment ions were observed at *m*/*z* 1279.5 [M_3Na_ − Na − NaSO_3_ + H]^−^, 1177.5 [M_3Na_ − Na − 2NaSO_3_ + 2H]^−^, 1001.4 [M_3Na_ − Na − 2NaSO_3_−C_7_H_12_O_5_ (MeGlc) + 2H]^−^, 941.4 [M_3Na_ − Na − 2NaSO_3_−C_7_H_12_O_5_ (MeGlc) − CH_3_COO + H]^−^, 751.3 [M_3Na_ − Na − NaHSO_4_ − C_32_H_47_O_5_ (Agl) + H]^−^, and 255.0 [M_3Na_ − Na − 2NaSO_3_ − C_33_H_47_O_5_ (Agl) − C_5_H_8_O_4_ (Xyl) − C_6_H_10_O_4_ (Qui) − C_5_H_8_O_4_ (Xyl) + 2H]^−^. The (+)ESI-MS/MS of **9** demonstrated the sequential loss by the [M_3Na_ + Na]^+^ ion at *m*/*z* 1427.4 of three sulfate groups (ion peaks at *m*/*z* 1325.4 [M_3Na_ + Na − NaSO_3_ + H]^+^, 1223.5 [M_3Na_ + Na − 2NaSO_3_ + 2H]^+^, and 1121.5 [M_3Na_ + Na − 3NaSO_3_ + 3H]^+^).

All these data indicate that quadrangularisoside D_1_ (**9**) is 3*β*-*O*-[4-*O*-sodium sulfate-3-*O*-methyl-*β*-d-glucopyranosyl-(1→3)-*β*-d-xylopyranosyl-(1→4)-3-*O*-sodium sulfate-*β*-d-quinovopyranosyl-(1→2)-4-*O*-sodium sulfate-*β*-d-xylopyranosyl]-16*β*-acetoxyholosta-7,25-diene.

The molecular formula of quadrangularisoside D_2_ (**10**) was determined to be C_53_H_79_O_30_S_3_Na_3_ from the [M_3Na_ − Na]**^−^** ion peak at *m*/*z* 1337.3625 (calc. 1337.3619), [M_3Na_ − 2Na]^2**−**^ ion peak at *m*/*z* 657.1881 (calc. 657.1863), and [M_3Na_ − 3Na]^3**−**^ ion peak at *m*/*z* 430.4633 (calc. 430.4612) in the (−)HR-ESI-MS as well as from the [M_3Na_ + Na]**^+^** ion peak at *m*/*z* 1383.3390 (calc. 1383.3404) in the (+)HR-ESI-MS. The aglycone of quadrangularisoside D_2_ (**10**) was identical to that of quadrangularisoside B_2_ (**5**) (Table 7), having holotoxinogenin as a triterpenoid nucleus [31].

The (−)ESI-MS/MS of **10** demonstrated the fragmentation of the [M_3Na_ − Na]^−^ ion at *m*/*z* 1381.4. The peaks of fragment ions were observed at *m*/*z* 1217.4 [M_3Na_ − Na − NaHSO_4_]^−^, 1059.4 [M_3Na_ − Na − C_7_H_11_O_8_SNa (MeGlcSO_3_Na)]^−^, 939.4 [M_3Na_ − Na − NaHSO_4_ − C_7_H_11_O_8_SNa (MeGlcSO_3_Na)]^−^, 533.1 [M_3Na_ − Na − 2NaSO_3_ − C_30_H_43_O_4_ (Agl) − C_5_H_8_O_4_ (Xyl) + H]^−^, 387.1 [M_3Na_ − Na − 2NaSO_3_ − C_30_H_43_O_4_ (Agl) − C_5_H_8_O_4_ (Xyl) − C_6_H_10_O_4_ (Qui) + H]^−^, and 255.0 [M_3Na_ − Na − 2NaSO_3_ − C_30_H_43_O_4_ (Agl) − C_5_H_8_O_4_ (Xyl) − C_6_H_10_O_4_ (Qui) − C_5_H_8_O_4_ (Xyl) + 2H]^−^, corroborating the identity of the carbohydrate chains of compounds **8**−**10**. The (+)ESI-MS/MS of **10** demonstrated the loss of two sulfate groups: ion peaks at *m*/*z* 1264.4 [M_3Na_ + Na − NaSO_4_]^+^ and 1143.4 [M_3Na_ + Na − 2NaSO_4_]^+^.

All these data indicate that quadrangularisoside D_2_ (**10**) is 3*β*-*O*-[4-*O*-sodium sulfate-3-*O*-methyl-*β*-d-glucopyranosyl-(1→3)-*β*-d-xylopyranosyl-(1→4)-3-*O*-sodium sulfate-*β*-d-quinovopyranosyl-(1→2)-4-*O*-sodium sulfate-*β*-d-xylopyranosyl]-16-ketoholosta-9(11),25-diene.

The molecular formula of quadrangularisoside D_3_ (**11**) was determined to be C_55_H_83_O_33_S_3_Na_3_ from the [M_3Na_ − Na]**^−^** ion peak at *m*/*z* 1413.3785 (calc. 1413.3780), [M_3Na_ − 2Na]^2**−**^ ion peak at *m*/*z* 695.1963 (calc. 695.1944), and [M_3Na_ − 3Na]^3**−**^ ion peak at *m*/*z* 455.8006 (calc. 455.7998) in the (−)HR-ESI-MS as well as from the [M_3Na_ + Na]**^+^** ion peak at *m*/*z* 1459.3534 (calc. 1459.3564) in the (+)HR-ESI-MS. The aglycone of quadrangularisoside D_3_ (**11**) was established to be 25-peroxy-16*β*-acetoxyholosta-7,23*E*-diene-3*β*-ol and was identical to the aglycone of quadrangularisoside A (**1**) (Table 2), which was deduced from the comparison of their ^13^C NMR spectra.

The (−)ESI-MS/MS of **11** demonstrated the fragmentation of the [M_3Na_ − Na]^−^ ion at *m*/*z* 1413.4. The peaks of fragment ions were observed at *m*/*z* 1261.4 [M_Na_ − Na − OOH − NaSO_4_]^−^, 1103.3 [M_Na_ − Na − OOH − C_7_H_11_O_8_SNa (MeGlcSO_3_Na) + H]^−^, 723.3 [M_3Na_ − Na − OOH −C_7_H_11_O_8_SNa (MeGlcSO_3_Na) − C_5_H_8_O_4_ (Xyl) − C_6_H_9_O_7_SNa (QuiSO_3_Na) + H]^−^, as well as the ion peaks at the same *m*/*z* as in the mass spectra of the other quadrangularisosides of the group D (**8**−**10**): 751.3 [M_3Na_ − Na − NaHSO_4_ − C_32_H_47_O_6_ (Agl)+ H]^−^, 387.1 [M_3Na_ − Na − 2NaSO_3_ − C_32_H_47_O_6_ (Agl) − C_5_H_8_O_4_ (Xyl) − C_6_H_10_O_4_ (Qui) + H]^−^, and 255.0 [M_3Na_ − Na − 2NaSO_3_ − C_32_H_47_O_6_ (Agl) − C_5_H_8_O_4_ (Xyl) − C_6_H_10_O_4_ (Qui) − C_5_H_8_O_4_ (Xyl) + H]^−^.

All these data indicate that quadrangularisoside D_3_ (**11**) is 3*β*-*O*-[4-*O*-sodium sulfate-3-*O*-methyl-*β*-d-glucopyranosyl-(1→3)-*β*-d-xylopyranosyl-(1→4)-3-*O*-sodium sulfate-*β*-d-quinovopyranosyl-(1→2)-4-*O*-sodium sulfate-*β*-d-xylopyranosyl]-25-peroxy-16*β*-acetoxyholosta-7,23*E*-diene.

The molecular formula of quadrangularisoside D_4_ (**12**) was determined to be the same as that for **11** (C_55_H_83_O_33_S_3_Na_3_) from the [M_3Na_ − Na]**^−^** ion peak at *m*/*z* 1413.3762 (calc. 1413.3780), [M_3Na_ − 2Na]^2**−**^ ion peak at *m*/*z* 695.1954 (calc. 695.1944), and [M_3Na_ − 3Na]^3**−**^ ion peak at *m*/*z* 455.8011 (calc. 455.7998) in the (−)HR-ESI-MS as well as from the [M_3Na_ + Na]**^+^** ion peak at *m*/*z* 1459.3512 (calc. 1459.3564) in the (+)HR-ESI-MS, indicating the isomerism of these glycosides. The comparison of the ^13^C NMR spectra of the aglycone parts of quadrangularisosides D_4_ (**12**) and A_1_ (**2**) showed their identity (Table 3). Thus, the aglycone of quadrangularisoside D_4_ (**12**) was established to be 24*S*-peroxy-16*β*-acetoxyholosta-7,25-diene-3*β*-ol.

The (−)ESI-MS/MS of **12** demonstrated the fragmentation of the [M_3Na_ − Na]^−^ ion at *m*/*z* 1413.4. The peaks of fragment ions were observed at *m*/*z* 999.4 [M_Na_ − Na − OOH−NaSO_3_ − C_7_H_11_O_8_SNa (MeGlcSO_3_Na)]^−^, 867.4 [M_Na_ − Na − OOH − NaSO_3_ − C_7_H_11_O_8_SNa (MeGlcSO_3_Na) − C_5_H_8_O_4_ (Xyl)]^−^, and 255.0 [M_3Na_ − Na − 2NaSO_3_ − C_32_H_47_O_6_ (Agl) − C_5_H_8_O_4_ (Xyl) − C_6_H_10_O_4_ (Qui) − C_5_H_8_O_4_ (Xyl) + H]^−^.

All these data indicate that quadrangularisoside D_4_ (**12**) is 3*β*-*O*-[4-*O*-sodium sulfate-3-*O*-methyl-*β*-d-glucopyranosyl-(1→3)-*β*-d-xylopyranosyl-(1→4)-3-*O*-sodium sulfate-*β*-d-quinovopyranosyl-(1→2)-4-*O*-sodium sulfate-*β*-d-xylopyranosyl]-24*S*-peroxy-16*β*-acetoxyholosta-7,25-diene.

In the ^1^H and ^13^C NMR spectra of the carbohydrate moiety of quadrangularisoside E (**13**), four characteristic doublets at δ_H_ 4.67–5.19 (*J* = 7.3–7.7 Hz) and corresponding to them four signals of anomeric carbons at δ_C_ 104.2–104.7 were indicative of a tetrasaccharide chain and *β*-configurations of glycosidic bonds (Table 12). The ^1^H,^1^H-COSY and 1D TOCSY spectra of **13** showed the signals of the isolated spin systems assigned to the xylose, quinovose, glucose, and 3-*O*-methylglucose residues. So, the monosaccharide composition of **13** was identical to that in the quadrangularisosides of group C (**6**, **7**). The comparison of their ^13^C NMR spectra showed the coincidence of the signals corresponding to the monosaccharide residues from the first to the third. The signals of terminal 3-*O*-methylglucose residue in the ^13^C NMR spectrum of **13** differed from the corresponding signals in the spectrum of **6**, **7** due to the attachment of the third sulfate group to C(4) MeGlc4, causing α- and β-shifting effects (δ_C_ 76.1 (C(4) MeGlc4); δ_C_ 85.2 (C(3) MeGlc4); δ_C_ 76.4 (C(5) MeGlc4)) in quadrangularisoside E (**13**). Actually, the signals of terminal sugar moieties in the spectra of quadrangularisosides E (**13**) and D (**8**) were coincident, corroborating the sulfation of C(4) MeGlc4 in glycoside **13**. Hence, compound **13** has a novel carbohydrate chain with an unusual position of the third sulfate group in the terminal sugar unit.

The molecular formula of quadrangularisoside E (**13**) was determined to be C_54_H_81_O_31_S_3_Na_3_ from the [M_3Na_ − Na]^−^ ion peak at *m*/*z* 1367.3739 (calc. 1367.3725), [M_3Na_ − 2Na]^2**−**^ ion peak at *m*/*z* 672.1941 (calc. 672.1916), and [M_3Na_ − 3Na]^3**−**^ ion peak at *m*/*z* 440.4674 (calc. 440.4647) in the (−)HR-ESI-MS as well as from the [M_3Na_ + Na]**^+^** ion peak at *m*/*z* 1413.3488 (calc. 1413.3509) in the (+)HR-ESI-MS. The aglycone of quadrangularisoside E (**13**) was identical to the aglycone of quadrangularisosides B_2_ (**5**) and D_2_ (**10**)—holotoxinogenin (Table 7).

The (−)ESI-MS/MS of **13** demonstrated the fragmentation of the [M_3Na_ − Na]^−^ ion at *m*/*z* 1367.4. The peaks of fragment ions were observed at *m*/*z* 1247.4 [M_Na_ − Na − NaHSO_4_]^−^, 1089.4 [M_Na_ − Na − C_7_H_11_O_8_SNa (MeGlcSO_3_Na)]^−^, 969.4 [M_3Na_ − Na − NaHSO_4_−C_7_H_11_O_8_SNa (MeGlcSO_3_Na)]^−^, 825.4 [M_3Na_ − Na − C_7_H_11_O_8_SNa(MeGlcSO_3_Na) − C_6_H_10_O_8_SNa(GlcSO_3_Na) + H]^−^, and 255.0 [M_3Na_ − Na − C_30_H_43_O_4_ (Agl) − C_5_H_7_O_7_SNa(XylSO_3_Na) − C_6_H_10_O_4_ (Qui) − C_6_H_9_O_8_SNa(GlcSO_3_Na) − H]^−^. As result of the fragmentation of the [M_3Na_ + Na]^+^ ion at *m*/*z* 1413.4 in the (+)ESI-MS/MS of **13**, the peaks of fragment ions were observed at *m*/*z* 963.5 [M_3Na_ + Na − C_30_H_43_O_3_ (Agl) + H]^+^ and 685.4 [M_3Na_ + Na − C_30_H_43_O_3_ (Agl) − C_7_H_11_O_8_SNa (MeGlcSO_3_Na) + H]^+^.

All these data indicate that quadrangularisoside E (**13**) is 3*β*-*O*-[4-*O*-sodium sulfate-3-*O*-methyl-*β*-d-glucopyranosyl-(1→3)-6-*O*-sodium sulfate-*β*-d-glucopyranosyl-(1→4)-*β*-d-quinovopyranosyl-(1→2)-4-*O*-sodium sulfate-*β*-d-xylopyranosyl]-16-ketoholosta-9(11),25-diene.

Thus, 13 unknown earlier triterpene glycosides were isolated from the Vietnamese sea cucumber *Colochirus quadrangularis*. The glycosides include five different carbohydrate chains (quadrangularisosides of the groups A–E) and seven holostane aglycones. The trisulfated carbohydrate chains of quadrangularisosides of the groups D and E are novel and characterized by the position of one of the sulfate groups at C(4) of the terminal 3-*O*-methylglucose residue, which is unusual for the glycosides from sea cucumbers. Two novel aglycones having hydroperoxyl groups in the side chains were discovered in quadrangularisosides A (**1**) and D_3_ (**11**) (hydroperoxyl group at C(25)) as well as in quadrangularisosides A_1_ (**2**) and D_4_ (**12**) (hydroperoxyl group at C(24)). The finding of such functionalities in the aglycones of triterpene glycosides from the sea cucumbers is rare, and two other cases were reported only last year [24,25].

As to the structures of the glycosides isolated earlier by the Chinese researchers from the same species [12,13,14,15], only two compounds—philinopsides A and F—were identified by us in the glycosidic fraction of *C. quadrangularis*. Other two compounds, isolated earlier—pentactasides B and C—are presumably the glycosides identical to quadrangularisosides B (**4**) and B_1_ (**5**), because their carbohydrate chain structures were established incorrectly due to inaccuracies made by the authors in the NMR data interpretation. The glycosides with di- (pentactaside III) and trisaccharide chains (pentactasides I and II) as well as pentactaside E, having 16-ketoholosta-7,25-diene-3*β*-ol as the aglycone, have not been found in the glycosidic sum obtained by us from *C. quadrangularis*.

### 2.3. Bioactivity of the Glycosides

The cytotoxic activities of compounds **1**–**13** as well as known earlier cladoloside C (used as positive control) against mouse erythrocytes (hemolytic activity), neuroblastoma Neuro 2a cells, and normal epithelial JB-6 cells are presented in Table 13. The erythrocytes were more sensitive (all of the compounds were rather strong hemolytics) to the action of the glycosides than Neuro 2a or JB-6 cells, but the structure–activity relationships (SARs) observed for the glycosides **1**–**13** were similar for three cell lines investigated. For instance, the quadrangularisosides of the groups B (**3**, **4**) and C (**6**, **7**), as well as quadrangularisosides D (**8**) and D_1_ (**9**) demonstrated high hemolytic action and slightly decreased cytotoxicity against Neuro 2a and JB-6 cells. These compounds contain holostane aglycones without hydroperoxyl groups and di- or trisulfated tetrasaccharide chains. Such a combination of structural features provides high membranolytic activity even if three sulfate groups are present.

The membranolytic activity of monosulfated quadrangularisosides of the group A (**1**, **2**) were decreased in comparison with that for the most active glycosides **3**, **4**, **6**–**9** due to the presence of the hydroperoxyl group. However, the combination of the aglycones with the hydroperoxy-group in the side chains and trisulfated carbohydrate chains, as in quadrangularisosides D_3_ (**11**) and D_4_ (**12**), significantly decreased the activity.

The unusual position of the third sulfate group at C-4 MeGlc4 in the carbohydrate chain of quadrangularisoside E (**13**) does not significantly influence the membranolytic activity, and the glycoside is rather strong hemolytic and cytotoxin.

The glycosides in this series having aglycones with a 7(8)-double bond and 16β-*O*-acetic group were more active in comparison with those having aglycones characterized by a 9(11)-double bond.

To detect the influence of compounds **1**–**13** on the cell viability, formation, and growth of colonies of human colorectal adenocarcinoma HT-29 cells, the cells were treated with various concentrations of the compounds (0–20 μM) for 24 h, and then cell viability was assessed by MTS (3-(4,5-dimethylthiazol-2-yl)-5-(3-carboxymethoxyphenyl)-2-(4-sulfophenyl)-2H-tetrazolium) assay. The concentrations of glycosides **1**–**13** that cause 50% of the inhibition of HT-29 cells viability are given in Table 13. It was shown that quadrangularisosides A (**1**), A_1_ (**2**), D_2_ (**10**), D_3_ (**11**), and D_4_ (**12**) are non-cytotoxic against HT-29 cells at the dose up to 20 μM. Compounds **6**, **7**, and **13** possess moderate cytotoxic activity, while the rest (**3**−**5**, **8**, **9**) effectively suppressed the cell viability of HT-29 cells (Table 13).

To investigate the effect of glycosides **1**–**13** on the colony formation of HT-29 cells, the concentrations lower than IC_50_ were chosen. The data concerning the inhibitory activity of the compounds on colony formation are presented as a concentration that causes 50% of the inhibition of colonies number (ICCF_50_) (Table 13). The glycosides having hydroperoxyl groups in the aglycones (**11**, **12**, as well as **1**) were shown to not inhibit the colony formation and growth of HT-29 cells for 50% under concentration even at 20 μM or demonstrated only a slight effect, correspondingly. Meanwhile, quadrangularisoside A_1_ (**2**) having this functionality was surprisingly active. Interestingly, quadrangularisosides C (**6**), C_1_ (**7**), and E (**13**) possessed strong inhibitory activity on colony formation in HT-29 cells at concentrations much lower that their IC_50_ values. This can indicate that their mechanism of action is related with the regulation of specific signaling pathways rather than a direct toxic effect. The structure–activity relationships (SARs) observed for HT-29 cells were largely similar with the SARs of the glycosides in relation to the other cell lines investigated. The main contributors influencing the SARs were the hydroperoxyl and sulfate groups presented in the glycosides. Generally, the quadrangularisosides of group B (disulfated compounds with the second sulfate group attached to C-3 Qui2 and without hydroperoxyls in the aglycones) were the most active in all tests. 

Initially, the effects of X-ray irradiation of 1 Gy or the individual compounds at a concentration of 0.02 μM on the colony formation of HT-29 cells were checked. None of the glycosides **1**–**13** influenced the process of colony formation and proliferation at a dose of 0.02 μM (data not shown). The number of colonies of HT-29 cells was found to be decreased by 32% after radiation exposure at a dose of 1 Gy. Noticeably, the synergic effects of the glycosides (0.02 μM) and radioactive irradiation (1 Gy) decreasing the number of colonies was observed. Quadrangularisosides A (**1**), B_1_ (**4**), and D_1_ (**9**) enhanced the effect of radiation by about 30%, while compounds **3**, **5**, **6**, and **8** enhanced the radiation effect by about 20%, and glycosides **2** and **10** enhanced the radiation effect by less than 10% (Figure 2).

## 3. Materials and Methods

### 3.1. General Experimental Procedures

Specific rotation, Perkin-Elmer 343 Polarimeter; NMR, Bruker Avance III 500 (Bruker BioSpin GmbH, Rheinstetten, Germany) (500.13/125.77 MHz) or Avance III 700 Bruker FT-NMR (Bruker BioSpin GmbH, Rheinstetten, Germany) (700.00/176.03 MHz) (^1^H/^13^C) spectrometers; ESI MS (positive and negative ion modes), Agilent 6510 Q-TOF apparatus, sample concentration 0.01 mg/mL; HPLC, Agilent 1100 apparatus with a differential refractometer; columns Supelcosil LC-Si (4.6 × 150 mm, 5 μm), Supelco Discovery HS F5-5 (10 × 250 mm, 5 μm).

### 3.2. Animals and Cells

Specimens of the sea cucumber *Colochirus quadrangularis* (family Cucumariidae; order Dendrochirotida) were collected on the coral reefs near the seashore of Vietnam in the South China Sea. Sampling was performed by scuba diving in July 2016 (collector T.N. Dautova) at a depth of 7–9 m. Sea cucumbers were identified by S. Sh. Dautov; voucher specimens are preserved in A.V. Zhirmunsky National Scientific Center of Marine Biology, Vladivostok, Russia.

CD-1 mice weighing 18–20 g were purchased from RAMS ‘Stolbovaya’ nursery (Russia) and kept at the animal facility in standard conditions. All experiments were conducted in compliance with all of the rules and international recommendations of the European Convention for the Protection of Vertebrate Animals Used for Experimental Studies.

Mouse epithelial JB-6 cells Cl 41-5a and mouse neuroblastoma cell line Neuro 2a (ATCC^®^ CCL-131) were purchased from ATCC (Manassas, VA, USA).

HT-29 cell line (ATCC# HTB-28) was cultured in McCoy’s 5A medium supplemented with 10% fetal bovine serum (FBS) and penicillin–streptomycin solution. Cells were maintained in a sterile environment and kept in an incubator at 5% CO_2_ and 37 °C to promote growth. HT-29 cells were sub-cultured every 3–4 days by their rinsing with phosphate-buffered saline (PBS), adding trypsin to detach the cells from the tissue culture flask, and transferring 10–20% of the harvested cells to a new flask containing fresh growth media.

### 3.3. Extraction and Isolation

The sea cucumbers were extracted twice with refluxing 60% EtOH. The extract was evaporated to dryness and dissolved in water followed by chromatography on a Polychrom-1 column (powdered Teflon, Biolar, Latvia). The glycosides were eluted with 50% EtOH and evaporated (2500 mg of crude glycoside sum were obtained). Its subsequent chromatography on an Si gel column with the gradient of solvent systems CHCl_3_/EtOH/H_2_O (100:100:17) followed by (100:125:25) as the mobile phase gave three fractions (I–III) containing different groups of glycosides. Fraction I (525 mg) was submitted to HPLC on a Supelco Discovery HS F5-5 column with MeOH/H_2_O/NH_4_OAc (1 M water solution) (70/28/2) as the mobile phase, which resulted in the isolation of six known earlier compounds (lefevreoside A_2_ (2.3 mg), philinopside A (29.8 mg), lefevreoside C (39.8 mg), neothyonidioside (25 mg), philinopside F (or violaceoside B) (8 mg), and colochiroside B_3_ (6.6 mg)) and two other subfractions (1.1 and 1.2). The HPLC of subfraction 1.2 on the same column but with CH_3_CN/H_2_O/NH_4_OAc (1 M water solution) (28/71/1) as the mobile phase gave 4.1 mg of known earlier colochiroside B_2_ and 16 mg of quadrangularisoside A (**1**). The rechromatography of the subfraction 1.1 under the same conditions gave 2.8 mg of quadrangularisoside A_1_ (**2**) and 1.8 mg of known colochiroside B_1_. The fraction II (709 mg) was submitted to HPLC on a Supelco Discovery HS F5-5 column with CH_3_CN/H_2_O/NH_4_OAc (1 M water solution) (35/63.5/1.5) as the mobile phase, which lead to the isolation of individual quadrangularisoside C_1_ (**7**) (3.8 mg) as well as two other subfractions (2.1 and 2.2). The subsequent HPLC of subfraction 2.2 on the same column but with MeOH/H_2_O/NH_4_OAc (67/31/2) as the mobile phase was followed by different ratios of the same solvents: (1) (75/22/3) gave 20.4 mg of quadrangularisoside B_1_ (**4**); (2) (66/32/2) gave 4.5 mg of quadrangularisoside C (**6**); (3) (63/35/2) gave 12.3 mg of quadrangularisoside B (**3**). The rechromatography of subfraction 2.1 on the silica-based column Supelcosil LC-Si with CHCl_3_/MeOH/H_2_O (65/25/2) as the mobile phase resulted in the isolation of 53 mg of known earlier hemioedemoside A and 30 mg of quadrangularisoside B_2_ (**5**). The most polar fraction III (195 mg) was submitted to HPLC on a Supelco Discovery HS F5-5 column with MeOH/H_2_O/NH_4_OAc (60/38.5/1.5) as the mobile phase to give four subfractions, 3.1–3.4. The subsequent HPLC of 3.4 on the same column with an MeOH/H_2_O/NH_4_OAc (58/40/2) solvent system resulted in the isolation of individual quadrangularisosides D (**8**) (6.2 mg) and D_1_ (**9**) (3.2 mg). Quadrangularisosides D_2_ (**10**) (3.4 mg) and E (**13**) (5.2 mg) were isolated by HPLC of subfraction 3.3 on a silica-based column Supelcosil LC-Si with CHCl_3_/MeOH/H_2_O (50/28/3) as the mobile phase followed by the rechromatography of two fractions obtained on a Supelco Discovery HS F5-5 column with MeOH/H_2_O/NH_4_OAc (60/38/2) as the mobile phase. The HPLC of subfraction 3.2 on a Supelco Discovery HS F5-5 column with MeOH/H_2_O/NH_4_OAc (60/37/3) gave 2.2 mg of quadrangularisoside D_3_ (**11**). The HPLC of subfraction 3.1 on the same column with MeOH/H_2_O/NH_4_OAc (62/35/3) gave 2.3 mg of quadrangularisoside D_4_ (**12**).

#### 3.3.1. Quadrangularisoside A (**1**)

Colorless powder; [α]_D_^20^ − 16 (*c* 0.1, 50% MeOH). NMR: See Table 1 and Table 2. (−)HR-ESI-MS *m*/*z*: 1209.5004 (calc. 1209.5004) [M_Na_ − Na]**^−^**; (+)HR-ESI-MS *m*/*z*: 1255.4779 (calc. 1255.4789) [M_Na_ + Na]^+^; (+)ESI-MS/MS *m*/*z*: 1223.5 [M_Na_ + Na − OOH + H]^+^, 1103.5 [M_Na_ + Na − OOH − NaSO_4_ + H]^+^, 927.5 [M_Na_ + Na− OOH − NaSO_4_ − C_7_H_12_O_5_ (MeGlc) + H]^+^, 795.4 [M_Na_ + Na− OOH − NaSO_4_ − C_7_H_12_O_5_ (MeGlc) − C_5_H_8_O_4_ (Xyl) + H]^+^, 729.2 [M_Na_ + Na− C_32_H_47_O_6_ (Agl) + H]^+^, 649.3 [M_Na_ + Na− OOH − NaSO_4_ − C_7_H_12_O_5_ (MeGlc) − C_5_H_8_O_4_ (Xyl) − C_6_H_10_O_4_ (Qui) + H]^+^, 609.2 [M_Na_ + Na− C_32_H_47_O_6_ (Agl) − NaHSO_4_]^+^, 477.1 [M_Na_ + Na− C_32_H_47_O_6_ (Agl) − NaHSO_4_ − C_5_H_8_O_4_ (Xyl) + H]^+^.

#### 3.3.2. Quadrangularisoside A_1_ (**2**)

Colorless powder; [α]_D_^20^ − 25 (*c* 0.1, 50% MeOH). NMR: See Table 1 and Table 3. (−)HR-ESI-MS *m*/*z*: 1209.5006 (calc. 1209.5004) [M_Na_ − Na]^−^; (+)HR-ESI-MS *m*/*z*: 1255.4772 (calc. 1255.4789) [M_Na_ + Na]^+^; (+)ESI-MS/MS *m*/*z*: 1237.5 [M_Na_ + Na − H_2_O]^+^, 1177.4 [M_Na_ + Na − H_2_O − CH_3_COOH]^+^, 1117.5 [M_Na_ + Na − H_2_O− NaHSO_4_]^+^, 729.2 [M_Na_ + Na− C_32_H_47_O_6_ (Agl) + H]^+^, 609.2 [M_Na_ + Na− C_32_H_47_O_6_ (Agl) − NaHSO_4_]^+^, 477.1 [M_Na_ + Na − C_32_H_47_O_6_ (Agl) − NaHSO_4_ − C_5_H_8_O_4_ (Xyl) + H]^+^.

#### 3.3.3. Quadrangularisoside B (**3**)

Colorless powder; [α]_D_^20^ − 22 (*c* 0.1, 50% MeOH). NMR: See Table 4 and Table 5. (−)HR-ESI-MS *m*/*z*: 1279.4489 (calc. 1279.4494) [M_2Na_ − Na]**^−^**, 628.2311 (calc. 628.2301) [M_2Na_ − 2Na]^2**−**^; (+)HR-ESI-MS *m*/*z*: 1325.4272 (calc. 1325.4278) [M_2Na_ + Na]**^+^**; (−)ESI-MS/MS *m*/*z*: 1219.4 [M_2Na_ − Na − CH_3_COOH]^−^, 1177.5 [M_2Na_ − Na − NaSO_3_ + H]^−^; (+)ESI-MS/MS *m*/*z*:1223.5 [M_2Na_ + Na − NaSO_3_ + H]^+^, 915.4 [M_2Na_ + Na − NaSO_3_ − C_7_H_12_O_5_ (MeGlc) − C_5_H_8_O_4_ (Xyl) + H]^+^, 813.4 [M_2Na_ + Na − C_32_H_47_O_5_ (Agl) − H]^+^, 711.1 [M_2Na_ + Na − C_32_H_47_O_5_ (Agl) − SO_3_Na]^+^, 579.1 [M_2Na_ + Na − C_32_H_47_O_5_ (Agl) − C_5_H_7_O_7_SNa (XylSO_3_Na) − H]^+^, 535.3 [M_2Na_ + Na − C_32_H_47_O_5_ (Agl) − SO_3_Na − C_7_H_12_O_5_ (MeGlc)]^+^, 403.0 [M_2Na_ + Na − C_32_H_47_O_5_ (Agl) − SO_3_Na − C_7_H_12_O_5_ (MeGlc) − C_5_H_8_O_4_ (Xyl)]^+^, 331.1 [M_2Na_ + Na − C_32_H_47_O_5_ (Agl) − C_5_H_7_O_7_SNa (XylSO_3_Na) − C_6_H_9_O_7_SNa (QuiSO_3_Na) − H]^+^.

#### 3.3.4. Quadrangularisoside B_1_ (**4**)

Colorless powder; [α]_D_^20^ − 23 (*c* 0.1, 50% MeOH). NMR: See Table 4 and Table 6. (−)HR-ESI-MS *m*/*z*: 1279.4502 (calc. 1279.4494) [M_2Na_ − Na]**^−^**, 628.2320 (calc. 628.2301) [M_2Na_ − 2Na]^2**−**^; (+)HR-ESI-MS *m*/*z*: 1325.4272 (calc. 1325.4278) [M_2Na_ + Na]**^+^**; (+)ESI-MS/MS *m*/*z*: 1223.5 [M_2Na_ + Na − NaSO_3_ + H]^+^, 1205.5 [M_2Na_ + Na − NaHSO_4_]^+^, 1085.5 [M_2Na_ + Na − 2NaHSO_4_]^+^, 897.4 [M_2Na_ + Na − NaHSO_4_ − C_7_H_12_O_5_ (MeGlc) − C_5_H_8_O_4_ (Xyl)]^+^, 711.1 [M_2Na_ + Na − C_32_H_47_O_5_ (Agl) − SO_3_Na]^+^, 579.1 [M_2Na_ + Na − C_32_H_47_O_5_ (Agl) − C_5_H_7_O_7_SNa (XylSO_3_Na) − H]^+^, 403.0 [M_2Na_ + Na − C_32_H_47_O_5_ (Agl) − SO_3_Na − C_7_H_12_O_5_ (MeGlc) − C_5_H_8_O_4_ (Xyl)]^+^, 331.1 [M_2Na_ + Na − C_32_H_47_O_5_ (Agl) − C_5_H_7_O_7_SNa (XylSO_3_Na) − C_6_H_9_O_7_SNa (QuiSO_3_Na) − H]^+^.

#### 3.3.5. Quadrangularisoside B_2_ (**5**)

Colorless powder; [α]_D_^20^ − 55 (*c* 0.1, 50% MeOH). NMR: See Table 4 and Table 7. (−)HR-ESI-MS *m*/*z*: 1235.4243 (calc. 1235.4232) [M_2Na_ − Na]**^−^**, 606.2189 (calc. 606.2170) [M_2Na_ − 2Na]^2**−**^; (+)HR-ESI-MS *m*/*z*: 1281.4004 (calc. 1281.4016) [M_2Na_ + Na]^+^; (+)ESI-MS/MS *m*/*z*: 1179.4 [M_2Na_ + Na − NaSO_3_ + H]^+^, 1059.5 [M_2Na_ + Na − NaSO_3_ − NaHSO_4_ + H]^+^, 711.1 [M_2Na_ + Na − C_30_H_43_O_4_ (Agl) − SO_3_Na]^+^, 579.1 [M_2Na_ + Na − C_30_H_43_O_4_ (Agl) − C_5_H_7_O_7_SNa (XylSO_3_Na) − H]^+^, 403.0 [M_2Na_ + Na − C_30_H_43_O_4_ (Agl) − SO_3_Na − C_7_H_12_O_5_ (MeGlc) − C_5_H_8_O_4_ (Xyl)]^+^, 331.1 [M_2Na_ + Na − C_30_H_43_O_4_ (Agl) − C_5_H_7_O_7_SNa (XylSO_3_Na) − C_6_H_9_O_7_SNa (QuiSO_3_Na) − H]^+^.

#### 3.3.6. Quadrangularisoside C (**6**)

Colorless powder; [α]_D_^20^ − 8 (*c* 0.1, 50% MeOH). NMR: See Table 8 and Table 9. (−)HR-ESI-MS *m*/*z*: 1309.4608 (calc. 1309.4599) [M_2Na_ − Na]**^−^**, 643.2369 (calc. 643.2354) [M_2Na_ − 2Na]^2**−**^; (+)HR-ESI-MS *m*/*z*: 1355.4373 (calc. 1355.4384) [M_2Na_ + Na]^+^; (−)ESI-MS/MS *m*/*z*: 1249.4 [M_2Na_ − Na − CH_3_COOH]^−^, 1189.5 [M_2Na_ − Na − NaHSO_4_]^−^, 1129.5 [M_2Na_ − Na − CH_3_COOH − NaHSO_4_]^−^, 677.2 [M_2Na_ − Na − NaHSO_4_ − C_32_H_47_O_5_ (Agl) − H]^−^, 563.1 [M_2Na_ − Na − NaHSO_4_ − C_32_H_47_O_5_ (Agl) − C_5_H_7_O_3_ (Xyl)]^−^, 417.1 [M_2Na_ − Na − NaHSO_4_ − C_32_H_47_O_5_ (Agl) − C_5_H_7_O_3_ (Xyl) − C_6_H_10_O_4_ (Qui)]^−^, 241.0 [M_2Na_ − Na − NaHSO_4_ − C_32_H_47_O_5_ (Agl) − C_5_H_7_O_3_ (Xyl) − C_6_H_10_O_4_ (Qui) − C_7_H_12_O_5_ (MeGlc)]^−^.

#### 3.3.7. Quadrangularisoside C_1_ (**7**)

Colorless powder; [α]_D_^20^ − 10 (*c* 0.1, 50% MeOH). NMR: See Table 8 and Table 10. (−)HR-ESI-MS *m*/*z*: 1311.4763 (calc. 1311.4756) [M_2Na_ − Na]**^−^**, 644.2447 (calc. 644.2432) [M_2Na_ − 2Na]^2**−**^; (+)HR-ESI-MS *m*/*z*: 1357.4520 (calc. 1357.4540) [M_2Na_ + Na]**^+^**; (−)ESI-MS/MS *m*/*z*:797.1 [M_2Na_ − Na − − C_32_H_49_O_5_ (Agl) − H]^−^, 677.2 [M_2Na_ − Na − NaHSO_4_ − C_32_H_49_O_5_ (Agl) − H]^−^, 563.1 [M_2Na_ − Na − NaHSO_4_ − C_32_H_49_O_5_ (Agl) − C_5_H_7_O_3_ (Xyl)]^−^, 417.1 [M_2Na_ − Na − NaHSO_4_ − C_32_H_49_O_5_ (Agl) − C_5_H_7_O_3_ (Xyl) − C_6_H_10_O_4_ (Qui)]^−^, 241.0 [M_2Na_ − Na − NaHSO_4_ − C_32_H_49_O_5_ (Agl) − C_5_H_7_O_3_ (Xyl) − C_6_H_10_O_4_ (Qui) − C_7_H_12_O_5_ (MeGlc)]^−^.

#### 3.3.8. Quadrangularisoside D (**8**)

Colorless powder; [α]_D_^20^ − 23 (*c* 0.1, 50% MeOH). NMR: See Table 5 and Table 11. (−)HR-ESI-MS *m*/*z*: 1381.3888 (calc. 1381.1881) [M_3Na_ − Na]**^−^**, 679.2012 (calc. 679.1995) [M_3Na_ − 2Na]^2**−**^, 445.1387 (calc. 445.1366) [M_3Na_ − 3Na]^3**−**^; (+)HR-ESI-MS *m*/*z*: 1427.3663 (calc. 1427.3666) [M_3Na_ + Na]**^+^**; (−)ESI-MS/MS *m*/*z*: 1261.4 [M_3Na_ − Na − NaHSO_4_]^−^, 1103.4 [M_3Na_ − Na − C_7_H_11_O_8_SNa (MeGlcSO_3_Na)]^−^, 751.3 [M_3Na_ − Na − NaHSO_4_ − C_32_H_47_O_5_ (Agl)+ H]^−^, 533.1 [M_3Na_ − Na − 2NaSO_3_ − C_33_H_47_O_5_ (Agl) − C_5_H_8_O_4_ (Xyl) + H]^−^, 387.1 [M_3Na_ − Na − 2NaSO_3_ − C_33_H_47_O_5_ (Agl) − C_5_H_8_O_4_ (Xyl) − C_6_H_10_O_4_ (Qui) + H]^−^, 255.0 [M_3Na_ − Na − 2NaSO_3_ − C_33_H_47_O_5_ (Agl) − C_5_H_8_O_4_ (Xyl) − C_6_H_10_O_4_ (Qui) − C_5_H_8_O_4_ (Xyl) + 2H]^−^.

#### 3.3.9. Quadrangularisoside D_1_ (**9**)

Colorless powder; [α]_D_^20^ − 26 (*c* 0.1, 50% MeOH). NMR: See Table 6 and Table 11. (−)HR-ESI-MS *m*/*z*: 1381.3891 (calc. 1381.3881) [M_3Na_ − Na]**^−^**, 679.2015 (calc. 679.1995) [M_3Na_ − 2Na]^2**−**^, 445.1390 (calc. 445.1366) [M_3Na_ − 3Na]^3**−**^; (+)HR-ESI-MS *m*/*z*: 1427.3653 (calc. 1427.3666) [M_3Na_ + Na]**^+^**; (−)ESI-MS/MS *m*/*z*: 1279.5 [M_3Na_ − Na − NaSO_3_ + H]^−^, 1177.5 [M_3Na_ − Na − 2NaSO_3_ + 2H]^−^, 1001.4 [M_3Na_ − Na − 2NaSO_3_ − C_7_H_12_O_5_ (MeGlc) + 2H]^−^, 941.4 [M_3Na_ − Na − 2NaSO_3_− C_7_H_12_O_5_ (MeGlc) − CH_3_COO + H]^−^, 751.3 [M_3Na_ − Na − NaHSO_4_ − C_32_H_47_O_5_ (Agl)+ H]^−^, 255.0 [M_3Na_ − Na − 2NaSO_3_ − C_33_H_47_O_5_ (Agl) − C_5_H_8_O_4_ (Xyl) − C_6_H_10_O_4_ (Qui) − C_5_H_8_O_4_ (Xyl) + 2H]^−^; (+)ESI-MS/MS *m*/*z*: 1325.4 [M_3Na_ + Na − NaSO_3_ + H]^+^, 1223.5 [M_3Na_ + Na − 2NaSO_3_ + 2H]^+^ and 1121.5 [M_3Na_ + Na − 3NaSO_3_ + 3H]^+^.

#### 3.3.10. Quadrangularisoside D_2_ (**10**)

Colorless powder; [α]_D_^20^ − 33 (*c* 0.1, 50% MeOH). NMR: See Table 7 and Table 11. (−)HR-ESI-MS *m*/*z*: 1337.3625 (calc. 1337.3619) [M_3Na_ −Na]**^−^**, 657.1881 (calc. 657.1863) [M_3Na_ − 2Na]^2**−**^, 430.4633 (calc. 430.4612) [M_3Na_ − 3Na]^3**−**^; (+)HR-ESI-MS *m*/*z*: 1383.3390 (calc. 1383.3404) [M_3Na_ + Na]**^+^**; (−)ESI-MS/MS *m*/*z*: 1217.4 [M_3Na_ − Na − NaHSO_4_]^−^, 1059.4 [M_3Na_ − Na − C_7_H_11_O_8_SNa (MeGlcSO_3_Na)]^−^, 939.4 [M_3Na_ − Na − NaHSO_4_ − C_7_H_11_O_8_SNa (MeGlcSO_3_Na)]^−^, 533.1 [M_3Na_ − Na − 2NaSO_3_ − C_30_H_43_O_4_ (Agl) − C_5_H_8_O_4_ (Xyl) + H]^−^, 387.1 [M_3Na_ − Na − 2NaSO_3_ − C_30_H_43_O_4_ (Agl) − C_5_H_8_O_4_ (Xyl) − C_6_H_10_O_4_ (Qui) + H]^−^, 255.0 [M_3Na_ − Na − 2NaSO_3_ − C_30_H_43_O_4_ (Agl) − C_5_H_8_O_4_ (Xyl) − C_6_H_10_O_4_ (Qui) − C_5_H_8_O_4_ (Xyl) + 2H]^−^; (+)ESI-MS/MS *m*/*z*: 1264.4 [M_3Na_ + Na − NaSO_4_]^+^, 1143.4 [M_3Na_ + Na − 2NaSO_4_]^+^.

#### 3.3.11. Quadrangularisoside D_3_ (**11**)

Colorless powder; [α]_D_^20^ − 19 (*c* 0.1, 50% MeOH). NMR: See Table 2 and Table 11. (−)HR-ESI-MS *m*/*z*: 1413.3785 (calc. 1413.3780) [M_3Na_ − Na]**^−^**, 695.1963 (calc. 695.1944) [M_3Na_ − 2Na]^2**−**^, 455.8006 (calc. 455.7998) [M_3Na_ − 3Na]^3**−**^; (+)HR-ESI-MS *m*/*z*: 1459.3534 (calc. 1459.3564) [M_3Na_ + Na]**^+^**; (−)ESI-MS/MS *m*/*z*: 1261.4 [M_Na_ − Na − OOH − NaSO_4_]^−^, 1103.3 [M_Na_ − Na − OOH − C_7_H_11_O_8_SNa (MeGlcSO_3_Na) + H]^−^, 723.3 [M_3Na_ − Na − OOH − C_7_H_11_O_8_SNa (MeGlcSO_3_Na) − C_5_H_8_O_4_ (Xyl) − C_6_H_9_O_7_SNa (QuiSO_3_Na) + H]^−^,751.3 [M_3Na_ − Na − NaHSO_4_ − C_32_H_47_O_6_ (Agl)+ H]^−^, 387.1 [M_3Na_ − Na − 2NaSO_3_ − C_32_H_47_O_6_ (Agl) − C_5_H_8_O_4_ (Xyl) − C_6_H_10_O_4_ (Qui) + H]^−^, 255.0 [M_3Na_ − Na − 2NaSO_3_ − C_32_H_47_O_6_ (Agl) − C_5_H_8_O_4_ (Xyl) − C_6_H_10_O_4_ (Qui) − C_5_H_8_O_4_ (Xyl) + H]^−^.

#### 3.3.12. Quadrangularisoside D_4_ (**12**)

Colorless powder; [α]_D_^20^ − 25 (*c* 0.1, 50% MeOH). NMR: See Table 3 and Table 11. (−)HR-ESI-MS *m*/*z*: 1413.3762 (calc. 1413.3780) [M_3Na_ − Na]**^−^**, 695.1954 (calc. 695.1944) [M_3Na_ − 2Na]^2**−**^, 455.8011 (calc. 455.7998) [M_3Na_ − 3Na]^3**−**^; (+)HR-ESI-MS *m*/*z*: 1459.3512 (calc. 1459.3564)[M_3Na_ + Na]**^+^**; (−)ESI-MS/MS *m*/*z*: 999.4 [M_Na_ − Na − OOH − NaSO_3_ − C_7_H_11_O_8_SNa (MeGlcSO_3_Na)]^−^, 867.4 [M_Na_ − Na − OOH − NaSO_3_ − C_7_H_11_O_8_SNa (MeGlcSO_3_Na) − C_5_H_8_O_4_ (Xyl)]^−^, 255.0 [M_3Na_ − Na − 2NaSO_3_ − C_32_H_47_O_6_ (Agl) − C_5_H_8_O_4_ (Xyl) − C_6_H_10_O_4_ (Qui) − C_5_H_8_O_4_ (Xyl) + H]^−^.

#### 3.3.13. Quadrangularisoside E (**13**)

Colorless powder; [α]_D_^20^ − 38 (*c* 0.1, 50% MeOH). NMR: See Table 7 and Table 12. (−)HR-ESI-MS *m*/*z*: 1367.3739 (calc. 1367.3725) [M_3Na_ − Na]**^−^**, 672.1941 (calc. 672.1916) [M_3Na_ − 2Na]^2**−**^, 440.4674 (calc. 440.4647) [M_3Na_ − 3Na]^3**−**^; (+)HR-ESI-MS *m*/*z*: 1413.3488 (calc. 1413.3509) [M_3Na_ + Na]**^+^**; (−)ESI-MS/MS *m*/*z*: 1247.4 [M_Na_ − Na − NaHSO_4_]^−^, 1089.4 [M_Na_ − Na − C_7_H_11_O_8_SNa (MeGlcSO_3_Na)]^−^, 969.4 [M_3Na_ − Na − NaHSO_4_ − C_7_H_11_O_8_SNa (MeGlcSO_3_Na)]^−^, 825.4 [M_3Na_ − Na − C_7_H_11_O_8_SNa (MeGlcSO_3_Na) − C_6_H_10_O_8_SNa (GlcSO_3_Na) + H]^−^, 255.0 [M_3Na_ − Na − C_30_H_43_O_4_ (Agl) − C_5_H_7_O_7_SNa (XylSO_3_Na) − C_6_H_10_O_4_ (Qui) − C_6_H_9_O_8_SNa (GlcSO_3_Na) − H]^−^; (+)ESI-MS/MS m/z: 963.5 [M_3Na_ + Na − C_30_H_43_O_3_ (Agl) + H]^+^, 685.4 [M_3Na_ + Na − C_30_H_43_O_3_ (Agl) − C_7_H_11_O_8_SNa (MeGlcSO_3_Na) + H]^+^.

### 3.4. Cytotoxic Activity (MTT Assay)

All compounds were tested in concentrations from 1.5 μM to 100 μM using two-fold dilution in dH_2_O. The solutions (20 µL) of tested substances in different concentrations and cell suspension (180 µL) were added in wells of 96-well plates (1 × 10^4^ cells/well) and incubated 24 h at 37 °C and 5% CO_2_. After incubation, the medium with tested substances was replaced by 100 μL of fresh medium. Then, 10 μL of MTT (thiazoyl blue tertrazolium bromide) stock solution (5 mg/mL) was added to each well, and the microplate was incubated for 4 h. After that, 100 μL of SDS-HCl solution (1 g SDS/10 mL dH_2_O/17 μL 6 N HCl) was added to each well followed by incubation for 4–18 h. The absorbance of the converted dye formazan was measured using a Multiskan FC microplate photometer (Thermo Scientific, Waltham, MA, USA) at a wavelength of 570 nm. The cytotoxic activity of the substances was calculated as the concentration that caused 50% metabolic cell activity inhibition (IC_50_). All the experiments were made in triplicate, *p* < 0.05.

### 3.5. Cytotoxic Activity (MTS Assay)

HT-29 cells (1.0 × 10^4^/200 μL) were seeded in 96-well plates for 24 h at 37 °C in a 5% CO_2_ incubator. The cells were treated with compounds **1**–**13** at concentrations ranging from 0 to 20 μM for an additional 24 h. Subsequently, cells were incubated with 15 μL MTS reagent for 3 h, and the absorbance in each well was measured at 490/630 nm using a microplate reader “Power Wave XS” (Bio Tek, Winooski, VT, USA). All the experiments were repeated three times, and the mean absorbance values were calculated. The results are expressed as the percentage of inhibition that produced a reduction in absorbance by the compound’s treatment compared to the non-treated cells (control). All the experiments were made in triplicate, *p* < 0.01.

### 3.6. Hemolytic Activity

Blood was taken from CD-1 mice (18–20 g). Erythrocytes were isolated from the blood of albino CD-1 mice by centrifugation with phosphate-buffered saline (pH 7.4) during 5 minutes at 4 °C by 450 g on a centrifuge LABOFUGE 400R (Heraeus, Germany) for three times. Then, the residue of erythrocytes was resuspended in ice cold phosphate saline buffer (pH 7.4) to a final optical density of 1.5 at 700 nm and kept on ice [39]. For the hemolytic assay, 180 µL of erythrocyte suspension was mixed with 20 µL of test compound solution in V-bottom 96-well plates. After 1 h of incubation at 37 °C, plates were exposed to centrifugation 10 min at 900 g on a laboratory centrifuge LMC-3000 (Biosan, Riga, Latvia) [40]. Then, we carefully separated 100 µL of supernatant and transferred it onto new flat plates, respectively. The lysis of erythrocytes was determined by measuring of the concentration of hemoglobin in the supernatant with a microplate photometer Multiskan FC (Themo Scientific, USA), λ = 570 nm [41]. The effective dose causing 50% hemolysis of erythrocytes (ED_50_) was calculated using the computer program SigmaPlot 10.0. All the experiments were made in triplicate, *p* < 0.05.

### 3.7. Soft Agar Assay

HT-29 cells (2.4 × 10^4^/mL) were seeded into 6-well plates and treated with compounds **1**–**13** at a concentrations range 0–20 μM in 1 mL of 0.3% Basal Medium Eagle (BME) agar containing 10% FBS, 2 mM L-glutamine, and 25 μg/mL gentamicin. The cultures were maintained at 37 °C in a 5% CO_2_ incubator for 14 days, and the cell’s colonies were scored using a microscope “Motic AE 20” (Scientific Instrument Company, Campbell, CA, USA) and the Motic Image Plus (Scientific Instrument Company, Campbell, CA, USA) computer program.

### 3.8. Radiation Exposure

Irradiation was delivered at room temperature using 1 Gy of the X-ray system XPERT 80 (KUB Technologies, Inc., Milford, CT, USA). The absorber dose was measured using an X-ray radiation clinical dosimeter DRK-1 (Akselbant, Moscow, Russia).

### 3.9. Cell Irradiation

HT-29 cells (5.0 × 10^5^/5 mL) were plated at 60 mm dishes and incubated for 24 h. After the incubation, the cells were cultured in the presence or absence of 0.02 μM of compounds **1**–**13** for an additional 24 h before irradiation at the dose of 1 Gy. Immediately after irradiation, the cells were returned to the incubator for recovery. Three hours later, cells were harvested and used for soft agar assay to establish the synergism between irradiation and the investigated compounds.

## 4. Conclusions

It is interesting to note that the aglycones having both 7(8)- and 9(11)-double bonds are present in the glycosidic sum of *C. quadrangularis* whenever the majority of sea cucumber species usually biosynthesize the aglycones with the certain position of double bond in the triterpene nucleus. Only six species of the holothurians besides *C. quadrangularis* are known so far to produce simultaneously the aglycones with different positions of double bonds in the nuclei: *Cucumaria frondosa* [42], *Australostichopus mollis* [43], and *Colochirus robustus* [20]—the aglycones with 7(8)- and 9(11)-double bonds; *Synapta maculata*—the aglycones with 7(8)- and 8(9)-double bonds [44]; and *Psolus fabricii* and *Cucumaria fallax*—the aglycones with 7(8)-, 8(9)-, and 9(11)-double bonds [37,45].

This peculiarity is interesting from the viewpoint of the biosynthesis of triterpene precursors of these aglycones. The species of sea cucumbers selectively biosynthesizing the aglycones with a certain position of the intra-nucleus double bond and the other species that can produce the aglycones having different positions of the double bond probably have diversely functioning oxydosqualene cyclases—the enzymes cyclizing 2,3-oxidosqualene to different triterpene alcohols—precursors of the aglycones. The reasons for such differences between the organisms and the producers of the glycosides are still unclear and have to be studied.

Triterpene glycosides of the sea cucumbers are metabolites that are formed by the mosaic type of biosynthesis [30,37], i.e., the aglycones and carbohydrate moieties are biosynthesized simultaneously and independently from each other. One of the results of such biosynthesis is the appearance of the glycosides having sugar chains identical to each other but structurally diverse aglycones. Such glycosides are usually attributed to one group. The glycosidic composition of *C. quadrangularis* is not an exception, since the glycosides of five groups have been isolated: the quadrangularisosides of group A (with a tetrasaccharide monosulfated chain), group B (with tetrasaccharide disulfated chain having the second sulfate group at C(3)Qui2), group C (with a tetrasaccharide disulfated chain having the sulfated by C(6) glucose residue as the third unit), group D (with a tetrasaccharide trisulfated chain having the sulfate groups at C(4)Xyl1, C(3)Qui2, and C(4)MeGlc4) and finally group E (with a tetrasaccharide trisulfated chain having the sulfate groups at C(4)Xyl1, C(6)Glc3, and C(4)MeGlc4). The majority of known compounds found in *C. quadrangularis* have carbohydrate chains identical to that of the quadrangularisoside of group A. Among them are colochirosides B_1_–B_3_ [20], lefevreosides A_2_ and C [21], neothyonidioside [22], and philinopside A [12]. The biogenetic relationships between the groups of these glycosides are obvious (Figure 3): the carbohydrate chain of quadrangularisosides of group A is a biosynthetic precursor of the di- and trisulfated carbohydrate chains of groups B and D. The carbohydrate chain of the quadrangularisosides of group C is a biosynthetic precursor of the sugar chain of quadrangularisoside E.

The mosaicism of the biosynthetic network of the glycosides is also illustrated by the biogenetic relationships of the aglycone parts of their molecules (Figure 3). Thus, the same aglycones can be glycosylated by different carbohydrate moieties as in quadrangularisosides A (**1**) and D_3_ (**11**); A_1_ (**2**) and D_4_ (**12**); philinopside A [12], quadrangularisosides B (**3**) and D (**8**); lefevreoside C [21], quadrangularisosides B_1_ (**4**) and D_1_ (**9**); neothyonidioside [22], quadrangularisosides B_2_ (**5**), D_2_ (**10**), and E (**14**). It is peculiar that only quadrangularisoside C (**6**) has the aglycone that is not represented in the other groups of quadrangularisosides.

It is known that sea cucumber triterpene glycosides are taxonomically specific and can be used as chemotaxonomic markers for different systematic groups of holothuroids [7,8,9,10]. Since the systematic position of the species under investigation was revised more than once and it was classified earlier as representative of the genus *Pentacta*, it seems to be important to analyze the glycosides isolated from *C. quadrangularis* from the viewpoint of chemotaxonomy. Chemically studied species of the sea cucumbers systematically related to *C. quadrangularis* are *Colochirus robustus* (=*Pentacta robustus*) and *Pseudocolochirus violaceus* (=*Colochirus violaceus*). It is also interesting to compare the glycosides of *C. quadrangularis* with those isolated from *Plesiocolochirus australis* (=*Pentacta australis*, =*Colochirus australis*).

Colochirosides B_1_–B_3_ isolated initially from *Colochirus robustus* [20] were identified in the glycosidic fraction of *C. quadrangularis*. Another four glycosides common to these species of sea cucumbers are hemoiedemoside A [20,23], philinopside F (=violaceuside B) [13,20,46], lefevreoside F [20,21], and neothyonidioside [20,22]. All these data indicate the systematic closeness of *Colochirus robustus* and *Colochirus quadrangularis*, which share seven identical compounds having holostane aglycones and tetrasaccharide chains and the reasonableness of their assignment to one genus.

Philinopside F found earlier in *C. quadrangularis* [13] was also identified in *Pseudocolochirus violaceus* but was discussed as a new glycoside named violaceuside B [46]. This compound was also identified by us in the glycosidic sum of *C. quadrangularis*. These species of sea cucumbers also have glycosides that are common for both of them such as philinopsides A, E, and F [12,13,46] and lefevreoside C [21,30]. This indicates that *Pseudocolochirus violaceus* and *C. quadrangularis* are systematically close species.

On the other hand, the glycosides from *Plesiocolochirus australis* (=*Pentacta australis*) [47] were significantly different from those from *C. quadrangularis* both by the aglycones structures (non-holostane aglycones of ds-penaustrosides A and B) and by the carbohydrate chains structures (pentasaccharide branched chains in ds-penaustrosides A–D [47]). So, the exclusion of *C. quadrangularis* from the genus *Pentacta* seems accurate.

## Figures and Tables

**Figure 1 marinedrugs-18-00394-f001:**
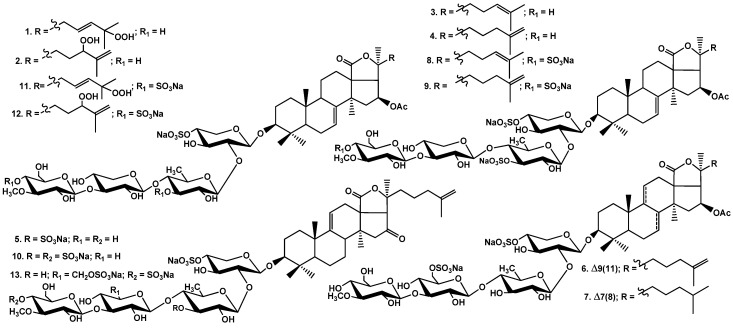
Chemical structures of glycosides isolated from *Colochirus quadrangularis:*
**1**—quadrangularisoside A; **2**—quadrangularisoside A_1_; **3**—quadrangularisoside B; **4**—quadrangularisoside B_1_; **5**—quadrangularisoside B_2_; **6**—quadrangularisoside C; **7**—quadrangularisoside C_1_; **8**—quadrangularisoside D; **9**—quadrangularisoside D_1_, **10**—quadrangularisoside D_2_; **11**—quadrangularisoside D_3_; **12**—quadrangularisoside D_4_; **13**—quadrangularisoside E.

**Figure 2 marinedrugs-18-00394-f002:**
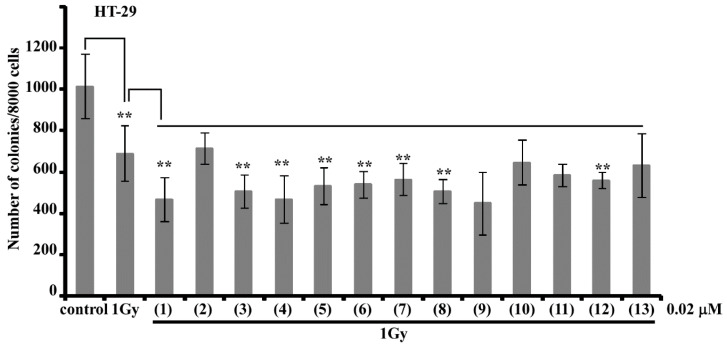
The effect of radioactive irradiation and a combination of radioactive irradiation and glycosides **1**–**13** on HT-29 cancer cells colony formation. HT-29 cells (8.0 × 10^3^) were cultured in the presence or absence of 0.02 μM compounds for an additional 24 h before irradiation at the dose of 1 Gy. Immediately after irradiation, cells were returned to the incubator for recovery. Three hours later, the cells were harvested and used for soft agar assay. Data are represented as the mean ± SD as determined from triplicate experiments. A Student’s *t*-test was used to evaluate the data with the following significance levels: ** *p* < 0.01.

**Figure 3 marinedrugs-18-00394-f003:**
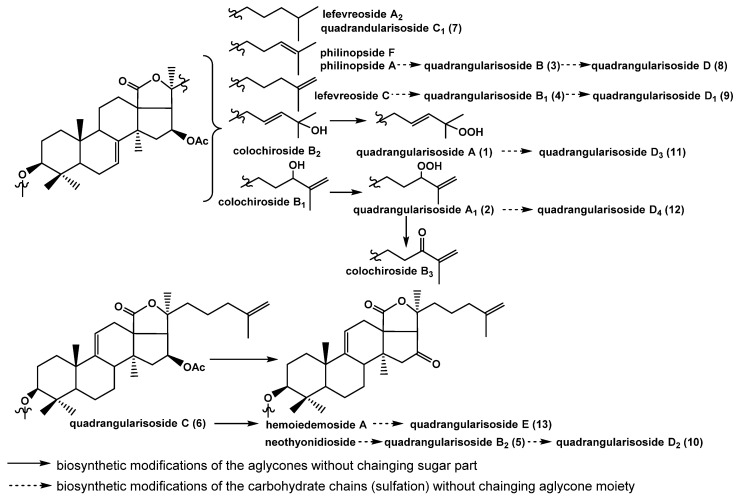
The biosynthetic network of triterpene glycosides from *C. quadrangularis*.

**Table 1 marinedrugs-18-00394-t001:** ^13^C and ^1^H NMR chemical shifts and HMBC and ROESY correlations of the carbohydrate moiety of quadrangularisosides A (**1**) and A_1_ (**2**). *^a^* Recorded at 176.03 MHz in C_5_D_5_N/D_2_O (4/1). *^b^* Bold = interglycosidic positions. *^c^* Italic = sulfate position. *^d^* Recorded at 700.00 MHz in C_5_D_5_N/D_2_O (4/1). Multiplicity by 1D TOCSY.

Atom.	δ_C_ mult. *^a,b,c^*	δ_H_ mult. *^d^* (*J* in Hz)	HMBC	ROESY
Xyl1 (1→C-3)				
1	104.8 CH	4.67 d (6.9)	C-3; C: 5 Xyl1	H-3; H-3, 5 Xyl1
2	**82.2** CH	3.99 t (8.9)	C: 1 Qui2; C: 1, 3 Xyl1	H-1 Qui2
3	75.0 CH	4.24 t (8.9)	C: 2, 4 Xyl1	H-1, 5 Xyl1
4	*76.2* CH	4.98 m	C: 3, 5 Xyl1	-
5	63.9 CH_2_	4.77 dd (5.4; 11.8)	C: 1, 3, 4 Xyl1	-
		3.73 dd (9.7; 11.5)	C: 1, 3, 4 Xyl1	H-1 Xyl1
Qui2 (1→2Xyl1)				
1	104.7 CH	4.99 d (7.8)	C: 2 Xyl1; C: 5 Qui2	H-2 Xyl1; H-3, 5 Qui2
2	75.8 CH	3.86 t (9.5)	C: 1, 3 Qui2	H-4 Qui2
3	74.8 CH	3.95 t (9.5)	C: 2, 4 Qui2	H-1, 5 Qui2
4	**85.6** CH	3.50 t (9.5)	C: 3, 5 Qui2, 1 Xyl3	H-1 Xyl3
5	71.4 CH	3.64 dd (6.1; 9.5)	-	H-1, 3 Qui2
6	17.8 CH_3_	1.60 d (6.1)	C: 4, 5 Qui2	H-4, 5 Qui2
Xyl3 (1→4Qui2)				
1	104.5 CH	4.77 d (7.8)	C: 4 Qui2	H-4 Qui2; H-3,5 Xyl3
2	73.4 CH	3.88 t (9.5)	C: 1, 3 Xyl3	-
3	**86.4** CH	4.11 m	C: 2, 4 Xyl3; 1 MeGlc4	H-1 MeGlc4; H-1, 5 Xyl3
4	68.7 CH	3.93 m	C: 5 Xyl3	-
5	65.9 CH_2_	4.11 dd (5.2; 10.4)	C: 3 Xyl3	-
		3.59 t (11.2)	C: 1, 3, 4 Xyl3	H-1, 3 Xyl3
MeGlc4 (1→3Xyl3)				
1	104.5 CH	5.20 d (8.0)	C: 3 Xyl3; C: 5 MeGlc4	H-3 Xyl3; H-3, 5 MeGlc4
2	74.5 CH	3.87 t (8.8)	-	-
3	87.0 CH	3.67 t (8.8)	C: 2, 4 MeGlc4, OMe	H-1, 5 MeGlc4; OMe
4	70.4 CH	3.88 t (8.8)	C: 5, 6 MeGlc4	-
5	77.5 CH	3.91 m	-	H-1 MeGlc4
6	61.8 CH_2_	4.36 dd (2.1; 11.7)	C: 4 MeGlc4	-
		4.04 dd (6.4; 11.9)	-	-
OMe	60.6 CH_3_	3.80 s	C: 3 MeGlc4	-

**Table 2 marinedrugs-18-00394-t002:** ^13^C and ^1^H NMR chemical shifts and HMBC and ROESY correlations of the aglycone moiety of quadrangularisosides A (**1**) and D_3_ (**11**). ^a^ Recorded at 176.03 MHz in C_5_D_5_N/D_2_O (4/1). ^b^ Recorded at 700.00 MHz in C_5_D_5_N/D_2_O (4/1).

Position	δ_C_ mult. ^a^	δ_H_ mult. (*J* in Hz) ^b^	HMBC	ROESY
1	35.8 CH_2_	1.33 m	-	H-19
		1.29 m	-	H-3, H-11
2	26.8 CH_2_	1.97 m	-	-
		1.79 m	-	H-19, H-30
3	89.1 CH	3.18 dd (4.3; 12.0)	C: 4, 30, 31, C:1 Xyl1	H-1, H-5, H-31, H-1Xyl1
4	39.3 C	-	-	-
5	47.9 CH	0.91 dd (4.3; 10.5)	C: 4, 6, 7, 9, 10, 19, 30, 31	H-3, H-31
6	23.2 CH_2_	1.94 m	C: 5, 7, 8, 10	H-19, H-30, H-31
7	120.3 CH	5.61 m	-	H-15, H-32
8	145.5 C	-	-	
9	47.0 CH	3.31 brd (14.3)	-	H-19
10	35.4 C	-	-	
11	22.3 CH_2_	1.71 m	-	H-1
		1.45 m	-	H-19
12	31.1 CH_2_	2.06 m	-	H-21
13	59.2 C	-	-	-
14	47.3 C	-	-	-
15	43.3 CH_2_	2.52 brdd (7.3; 12.4)	C: 13, 14, 16, 17, 32	H-7, H-32
		1.64 brd (7.3)	C: 14, 16, 32	-
16	74.8 CH	5.92 q (8.5)	C: 13, 15, 17, 20, OAc	H-32
17	54.3 CH	2.76 d (8.9)	C: 12, 13, 15, 18, 21	H-12, H-21, H-32
18	180.0 C	-	-	-
19	23.8 CH_3_	1.10 s	C: 5, 9, 10	H-2, H-6, H-9, H-11, H-30
20	84.8 C	-	-	-
21	28.2 CH_3_	1.53 s	C: 17, 20, 22	H-12, H-17, H-22, H-23
22	41.6 CH_2_	3.20 td (5.4; 14.0)	C: 20, 21, 23, 24	-
		2.66 dd (7.3; 14.0)	C: 17, 20, 21, 23, 24	H-21
23	124.2 CH	5.71 dt (6.6; 15.5)	C: 20, 22, 25	H-26, H-27
24	139.5 CH	5.97 d (15.8)	C: 22, 25, 26, 27	H-22, H-26, H-27
25	81.3 C	-	-	-
26	24.8 CH_3_	1.46 s	C: 23, 24, 25, 27	H-23, H-27
27	24.6 CH_3_	1.48 s	C: 23, 24, 25, 26	H-23, H-26
30	17.1 CH_3_	1.00 s	C: 3, 4, 5, 31	H-2, H-6, H-19, H-31
31	28.5 CH_3_	1.17 s	C: 3, 4, 5, 30	H-3, H-5, H-6, H-30, H-1 Xyl1
32	32.0 CH_3_	1.09 s	C: 8, 13, 14, 15	H-7, H-12, H-15, H-16, H-17
OCOCH_3_	170.7 C	-	-	-
OCOCH_3_	21.1 CH_3_	1.97 s	C: 16, OAc	-

**Table 3 marinedrugs-18-00394-t003:** ^13^C and ^1^H NMR chemical shifts and HMBC and ROESY correlations of the aglycone moiety of quadrangularisosides A_1_(**2**) and D_4_ (**11**). ^a^ Recorded at 176.03 MHz in C_5_D_5_N/D_2_O (4/1). ^b^ Recorded at 700.00 MHz in C_5_D_5_N/D_2_O (4/1).

Position	δ_C_ mult. ^a^	δ_H_ mult. (*J* in Hz) ^b^	HMBC	ROESY
1	35.9 CH_2_	1.32 m	-	H-3, H-5, H-11, H-19
2	26.8 CH_2_	1.97 m	-	-
		1.79 m	-	H-19, H-30
3	89.1 CH	3.18 dd (3.9; 12.0)	C:1 Xyl1	H-1, H-5, H-6, H-31, H-1Xyl1
4	39.3 C	-	-	-
5	47.9 CH	0.91 dd (4.6; 10.4)	C: 4, 6, 10, 19, 30	H-3, H-31
6	23.1 CH_2_	1.94 m	-	H-19, H-30, H-31
7	120.3 CH	5.60 m	-	H-15, H-32
8	145.7 C	-	-	-
9	47.1 CH	3.29 brd (14.5)	-	H-19
10	35.4 C	-	-	-
11	22.5 CH_2_	1.72 m	-	H-1
		1.46 m	-	-
12	31.3 CH_2_	2.10 m	-	-
13	59.4 C	-	-	-
14	47.1 C	-	-	-
15	43.8 CH_2_	2.58 brdd (7.3; 12.3)	C: 13, 17	H-7
		1.60 m	C: 17	H-12, H-32
16	75.5 CH	5.75 q (8.9)	-	H-21, H-32
17	54.6 CH	2.68 d (8.9)	C: 18	H-12, H-21, H-32
18	180.3 C	-	-	-
19	23.8 CH_3_	1.10 s	C: 5, 9, 10	H-1, H-2, H-6, H-9, H-30
20	85.3 C	-	-	-
21	27.9 CH_3_	1.52 s	C: 17, 20, 22	H-12, H-17, H-22, H-23
22	34.7 CH_2_	2.36 td (5.0; 13.3)	-	-
		2.19 td (4.2; 13.3)	-	H-21
23	26.8 CH_2_	1.85 m	-	H-21
		1.63 m	-	-
24	89.2 CH	4.51 t (6.2)	C: 26	H-22, H-26, H-27
25	144.9 C	-	-	-
26	113.6 CH_2_	5.14 brs	C: 24, 27	H-24
		5.02 brs	C: 24, 27	H-27
27	17.5 CH_3_	1.82 s	C: 24, 25, 26	H-26
30	17.2 CH_3_	1.00 s	C: 3, 4, 5, 31	H-2, H-6, H-19, H-31, H-6 Qui2
31	28.6 CH_3_	1.17 s	C: 3, 4, 5, 30	H-3, H-5, H-6, H-30, H-1 Xyl1
32	32.3 CH_3_	1.15 s	C: 8, 13, 14, 15	H-15, H-16, H-17
OCOCH_3_	171.0 C	-	-	-
OCOCH_3_	21.4 CH_3_	2.05 s	C: OAc	-

**Table 4 marinedrugs-18-00394-t004:** ^13^C and ^1^H NMR chemical shifts and HMBC and ROESY correlations of carbohydrate moiety of quadrangularisosides B (**3**), B_1_ (**4**), and B_2_ (**5**). *^a^* Recorded at 176.03 MHz in C_5_D_5_N/D_2_O (4/1). *^b^* Bold = interglycosidic positions. *^c^* Italic = sulfate position. *^d^* Recorded at 700.00 MHz in C_5_D_5_N/D_2_O (4/1). Multiplicity by 1D TOCSY.

Atom	δ_C_ mult. *^a,b,c^*	δ_H_ mult. *^d^* (*J* in Hz)	HMBC	ROESY
Xyl1 (1→C-3)				
1	104.6 CH	4.63 d (7.7)	C-3; C: 5 Xyl1	H-3; H-3, 5 Xyl1
2	**81.9** CH	3.96 dd (7.3; 8.9)	C: 1 Qui2; 1, 3 Xyl1	H-1 Qui2
3	75.3 CH	4.22 t (8.9)	C: 2, 4 Xyl1	H-1, 5 Xyl1
4	*76.0* CH	4.96 m	C: 3 Xyl1	-
5	64.1 CH_2_	4.79 dd (5.4; 11.6)	C: 1, 3, 4 Xyl1	-
		3.73 dd (9.3; 11.6)	C: 1, 4 Xyl1	H-1 Xyl1
Qui2 (1→2Xyl1)				
1	103.5 CH	5.10 d (7.1)	C: 2 Xyl1	H-2 Xyl1; H-3, 5 Qui2
2	75.7 CH	4.00 t (8.7)	C: 1, 3 Qui2	-
3	*81.5* CH	4.98 t (8.7)	C: 2, 4 Qui2	H-1, 5 Qui2
4	**79.2** CH	3.82 t (8.7)	C: 1 Xyl3	H-1 Xyl3
5	71.7 CH	3.65 dd (6.3; 8.7)	C: 1 Qui2	H-1 Qui2
6	18.3 CH_3_	1.53 d (5.7)	C: 4, 5 Qui2	H-4, 5 Qui2
Xyl3 (1→4Qui2)				
1	102.8 CH	5.09 d (6.8)	C: 4 Qui2; 5 Xyl3	H-4 Qui2; H-3,5 Xyl3
2	73.2 CH	4.10 t (9.1)	C: 1, 3 Xyl3	-
3	**86.0** CH	4.13 t (9.1)	C: 1 MeGlc4; 2, 4 Xyl3	H-1 MeGlc4
4	69.4 CH	4.03 m	C: 3 Xyl3	-
5	65.9 CH_2_	4.46 dd (5.3; 12.1)	C: 1, 3, 4 Xyl3	-
		3.77 dd (8.3; 12.1)	C: 1, 4 Xyl3	H-1 Xyl3
MeGlc4 (1→3Xyl3)				
1	104.3 CH	5.12 d (8.3)	C: 3 Xyl3	H-3 Xyl3; H-3, 5 MeGlc4
2	74.5 CH	3.85 t (8.3)	C: 1, 3 MeGlc4	-
3	86.8 CH	3.65 t (8.3)	C: 2, 4 MeGlc4, OMe	-
4	70.3 CH	3.86 t (8.3)	C: 5 MeGlc4	-
5	77.4 CH	3.86 m	C: 4 MeGlc4	H-1, 3 MeGlc4
6	61.8 CH_2_	4.33 d (12.8)	C: 4 MeGlc4	-
		4.00 dd (6.1; 12.1)	C: 5 MeGlc4	-
OMe	60.5 CH_3_	3.78 s	C: 3 MeGlc4	-

**Table 5 marinedrugs-18-00394-t005:** ^13^C and ^1^H NMR chemical shifts and HMBC and ROESY correlations of the aglycone moiety of quadrangularisosides B (**3**) and D (**8**).^a^ Recorded at 176.03 MHz in C_5_D_5_N/D_2_O (4/1). ^b^ Recorded at 700.00 MHz in C_5_D_5_N/D_2_O (4/1).

Atom	δ_C_ mult. *^a^*	δ_H_ mult. *^b^* (*J* in Hz)	HMBC	ROESY
1	35.8 CH_2_	1.32 m	-	-
		1.28 m	-	-
2	26.7 CH_2_	1.94 m	-	-
		1.76 m	-	H-19, H-30
3	89.2 CH	3.14 dd (4.2; 11.8)	C: 4, 30, 31, C:1 Xyl1	H-5, H-1Xyl1
4	39.3 C	-	-	-
5	47.8 CH	0.90 dd (4.5; 11.2)	C: 4, 6, 10, 30	H-1, H-3, H-31
6	23.1 CH_2_	1.93 m	-	H-31
7	120.3 CH	5.60 m	C: 9	H-15, H-32
8	145.6 C	-	-	-
9	47.0 CH	3.30 brd (16.5)	-	H-19
10	35.3 C	-	-	-
11	22.4 CH_2_	1.71 m	-	H-1
		1.47 m	-	-
12	31.2 CH_2_	2.09 m	C: 13, 14, 18	-
13	59.3 C	-	-	-
14	47.3 C	-	-	-
15	43.5 CH_2_	2.56 brdd (7.3; 12.2)	C: 13, 14, 17, 32	H-7, H-32
		1.60 brd (6.0)	-	-
16	75.2 CH	5.85 q (8.5)	C: 13, 15, 17, 20, OAc	H-32
17	54.3 CH	2.69 d (8.9)	C: 12, 13, 15, 18, 21	H-12, H-21, H-32
18	180.3 C	-	-	-
19	23.8 CH_3_	1.09 s	C: 1, 5, 9, 10	H-2, H-9, H-30
20	85.5 C	-	-	-
21	28.1 CH_3_	1.55 s	C: 17, 20, 22	H-12, H-17, H-23
22	38.5 CH_2_	2.42 dt (4.3; 12.3)	C: 20, 21	-
		1.82 dt (5.2; 12.3)	C: 17, 20, 21	-
23	23.5 CH	2.01 m	-	-
		1.92 m	-	-
24	123.9 CH	4.99 m	C: 26, 27	H-22, H-26
25	132.3 C	-	-	-
26	25.4 CH_3_	1.60 s	C: 24, 25, 27	H-24
27	17.6 CH_3_	1.54 s	C: 24, 25, 26	H-24
30	17.2 CH_3_	1.00 s	C: 3, 4, 5, 31	H-2, H-6, H-19, H-31
31	28.6 CH_3_	1.13 s	C: 3, 4, 5, 30	H-3, H-5, H-6, H-30, H-1 Xyl1
32	32.1 CH_3_	1.15 s	C: 8, 13, 14, 15	H-7, H-12, H-15, H-16, H-17
OCOCH_3_	170.9 C	-	-	-
OCOCH_3_	21.0 CH_3_	1.96 s	C: 16, OAc	-

**Table 6 marinedrugs-18-00394-t006:** ^13^C and ^1^H NMR chemical shifts and HMBC and ROESY correlations of the aglycone moiety of quadrangularisosides B_1_ (**4**) and D_1_ (**9**). ^a^ Recorded at 176.03 MHz in C_5_D_5_N/D_2_O (4/1). ^b^ Recorded at 700.00 MHz in C_5_D_5_N/D_2_O (4/1).

Atom	δ_C_ mult. ^a^	δ_H_ mult. ^b^ (*J* in Hz)	HMBC	ROESY
1	35.9 CH_2_	1.40 m	-	H-5, H-11, H-19
2	26.7 CH_2_	2.03 m	-	-
		1.95 m	-	-
3	89.2 CH	3.31 dd (4.2; 11.6)	-	H-5, H-31, H-1Xyl1
4	39.4 C	-	-	
5	47.8 CH	1.00 dd (5.0; 10.0)	C: 4, 10, 19, 30	H-1, H-3
6	23.1 CH_2_	2.00 m	-	H-30, H-31
7	120.2 CH	5.69 m	-	H-15, H-32
8	145.6 C	-	-	
9	47.0 CH	3.40 brd (14.0)	-	H-19
10	35.3 C		-	
11	22.4 CH_2_	1.81 m	-	H-1
		1.59 m	-	H-1
12	31.2 CH_2_	2.21 m	-	H-17, H-21
13	59.3 C	-	-	-
14	47.3 C	-	-	-
15	43.5 CH_2_	2.66 dd (7.4; 12.2)	-	H-7
		1.71m	-	-
16	75.3 CH	5.93brq (8.7)	-	H-32
17	54.5 CH	2.78 d (9.0)	C: 12, 13, 18, 21	H-12, H-21, H-32
18	180.3 C	-		
19	23.8 CH_3_	1.16 s	C: 1, 5, 9, 10	H-2, H-9
20	85.6 C	-		
21	28.1 CH_3_	1.62 s	C: 17, 20, 22	H-17, H-22
22	38.3 CH_2_	2.35 m	-	-
		1.91 m	-	-
23	22.9 CH_2_	1.56 m	-	-
		1.44 m	-	-
24	38.2 CH_2_	2.01 m	-	-
25	145.4 C	-	-	-
26	110.9 CH_2_	4.83brs	C: 24, 25, 27	
		4.84 brs	C: 24, 25, 27	
27	22.1 CH_3_	1.75 s	C: 24, 25, 26	
30	17.3 CH_3_	1.15 s	C: 3, 4, 5, 31	H-6
31	28.6 CH_3_	1.28 s	C: 3, 4, 5, 30	H-3, H-6
32	32.2 CH_3_	1.27 s	C: 8, 13, 14, 15	H-7, H-16, H-17
OCOCH_3_	170.8 C	-	-	-
OCOCH_3_	21.3 CH_3_	2.12 s	-	-

**Table 7 marinedrugs-18-00394-t007:** ^13^C and ^1^H NMR chemical shifts and HMBC and ROESY correlations of the aglycone moiety of quadrangularisosides B_2_ (**5**), D_2_ (**10**) and E (**13**). ^a^ Recorded at 176.03 MHz in C_5_D_5_N/D_2_O (4/1). ^b^ Recorded at 700.00 MHz in C_5_D_5_N/D_2_O (4/1).

Atom	δ_C_ mult. ^a^	δ_H_ mult. ^b^ (*J* in Hz)	HMBC	ROESY
1	36.0 CH_2_	1.69 m	C: 3	H-11, H-19
		1.28 m	C: 2	H-3
2	26.6 CH_2_	2.00 m	-	-
		1.80 brd (12.4)	-	H-19, H-30
3	88.8 CH	3.09 dd (4.2; 12.4)	C: 4, 30, 31, C:1 Xyl1	H-5, H-31, H-1 Xyl1
4	39.7 C	-	-	-
5	52.7 CH	0.78 brd (11.9)	C: 6, 10, 19, 30	H-3, H-7, H-31
6	20.9 CH_2_	1.60 m	-	H-31
		1.40 m	-	H-19, H-30
7	28.3 CH_2_	1.58 m	-	
		1.17 m	-	H-5, H-32
8	38.6 CH	3.12 brd (14.5)	-	H-15, H-19
9	150.9 C	-	-	-
10	39.4 C	-	-	-
11	111.1 CH	5.26 brd (4.7)	C: 8, 10, 12, 13	H-1
12	31.9 CH_2_	2.62 brd (18.1)	C: 9, 11, 13, 18	H-17, H-32
		2.46 dd (6.2; 18.1)	C: 9, 11, 13, 14, 18	H-21
13	55.8 C	-	-	-
14	41.9 C	-	-	-
15	51.9 CH_2_	2.38 d (15.6)	C: 13, 16, 17, 32	H-7, H-32
		2.09 d (15.6)	C: 8, 14, 16, 18, 32	H-8
16	214.4 C	-	-	-
17	61.2 CH	2.84 s	C: 12, 13, 16, 18, 20, 21	H-12, H-21, H-32
18	176.7 C	-	-	-
19	21.8 CH_3_	1.24 s	C: 1, 5, 9, 10	H-1, H-2, H-6, H-8, H-30
20	83.5 C	-	-	-
21	26.7 CH_3_	1.42 s	C: 17, 20, 22	H-12, H-17, H-23
22	38.1 CH_2_	1.71 m	C: 20, 21, 23, 24	H-21
		1.55 m	C: 21, 23	-
23	22.0 CH_2_	1.70 m	-	-
		1.43 m	-	-
24	37.7 CH_2_	1.88 dd (7.8; 15.0)	C: 22, 23, 25, 26	H-21, H-26
25	145.5 C	-	-	-
26	110.4 CH_2_	4.70 brs	C: 24, 27	H-27
		4.68 brs	C: 24, 27	H-24, H-27
27	22.1 CH_3_	1.62 s	C: 24, 25, 26	-
30	16.5 CH_3_	0.96 s	C: 3, 4, 5, 31	H-2, H-6, H-19, H-31, H-6 Qui2
31	27.9 CH_3_	1.10 s	C: 3, 4, 5, 30	H-3, H-5, H-6, H-30, H-1 Xyl1
32	20.5 CH_3_	0.88 s	C: 8, 13, 14, 15	H-7, H-12, H-15, H-17

**Table 8 marinedrugs-18-00394-t008:** ^13^C and ^1^H NMR chemical shifts and HMBC and ROESY correlations of carbohydrate moiety of quadrangularisosides C (**6**) and C_1_ (**7**). *^a^* Recorded at 176.03 MHz in C_5_D_5_N/D_2_O (4/1). *^b^* Bold = interglycosidic positions. *^c^* Italic = sulfate position. *^d^* Recorded at 700.00 MHz in C_5_D_5_N/D_2_O (4/1). Multiplicity by 1D TOCSY.

Atom	δ_C_ mult. *^a,b,c^*	δ_H_ mult. *^d^* (*J* in Hz)	HMBC	ROESY
Xyl1 (1→C-3)				
1	104.7 CH	4.67 d (7.3)	C-3	H-3; H-3, 5 Xyl1
2	**82.1** CH	3.95 t (7.3)	C: 1 Qui2; 1, 3 Xyl1	H-1 Qui2
3	74.6 CH	4.24 t (8.6)	C: 2, 4 Xyl1	H-1, 5 Xyl1
4	*76.1* CH	4.97 m	-	-
5	63.7 CH_2_	4.75 dd (4.9; 11.6)	C: 3, 4 Xyl1	-
		3.73 dd (9.8; 11.6)	-	H-1, 3 Xyl1
Qui2 (1→2Xyl1)				
1	104.5 CH	4.93 d (7.3)	C: 2 Xyl1	H-2 Xyl1; H-3, 5 Qui2
2	75.4 CH	3.87 t (9.2)	C: 1, 3 Qui2	H-4 Qui2
3	75.0 CH	3.92 t (9.2)	C: 2 Qui2	H-1, 5 Qui2
4	**86.9** CH	3.41 t (9.2)	C: 1 Glc3; 3, 5 Qui2	H-1 Glc3; H-2 Qui2
5	71.3 CH	3.60 dd (6.1; 9.2)	-	H-1, 3 Qui2
6	17.7 CH_3_	1.59 d (6.1)	C: 4, 5 Qui2	H-4, 5 Qui2
Glc3 (1→4Qui2)				
1	104.2 CH	4.75 d (7.6)	C: 4 Qui2	H-4 Qui2; H-3,5 Glc3
2	73.4 CH	3.85 t (8.6)	C: 1, 3 Glc3	-
3	**86.0** CH	4.18 t (8.6)	C: 1 MeGlc4; 2, 4 Glc3	H-1 MeGlc4, H-1 Glc3
4	69.2 CH	3.80 t (9.5)	C: 3, 5, 6 Glc3	H-6 Glc3
5	74.9 CH	4.10 m	-	H-1, 3 Glc3
6	*67.3* CH_2_	4.99 d (8.6)	C: 5 Glc3	-
		4.60 dd (6.7; 10.5)	-	H-4 Glc3
MeGlc4 (1→3Glc3)				
1	104.4 CH	5.21 d (7.9)	-	H-3 Glc3; H-3, 5 MeGlc4
2	74.5 CH	3.86 t (7.9)	C: 1, 3 MeGlc4	-
3	86.9 CH	3.67 t (8.8)	C: 2, 4 MeGlc4, OMe	H-1 MeGlc4
4	70.3 CH	3.90 m	C: 3, 5 MeGlc4	H-6 MeGlc4
5	77.5 CH	3.90 m	C: 6 MeGlc4	H-1, 3 MeGlc4
6	61.7 CH_2_	4.35 d (12.4)	-	-
		4.06 dd (7.1; 12.4)	C: 5 MeGlc4	-
OMe	60.6 CH_3_	3.80 s	C: 3 MeGlc4	-

**Table 9 marinedrugs-18-00394-t009:** ^13^C and ^1^H NMR chemical shifts and HMBC and ROESY correlations of the aglycone moiety of quadrangularisoside C (**6**) ^a^ Recorded at 176.03 MHz in C_5_D_5_N/D_2_O (4/1). ^b^ Recorded at 700.00 MHz in C_5_D_5_N/D_2_O (4/1).

Position	δ_C_ mult. ^a^	δ_H_ mult. (*J* in Hz) ^b^	HMBC	ROESY
1	36.1 CH_2_	1.69 m	-	H-11
		1.28 m	-	H-3, H-11
2	26.7 CH_2_	2.02 m	-	
		1.81 m	-	H-19, H-30
3	88.7 CH	3.11 dd (3.8; 11.5)	C: 30, 1 Xyl1	H-1, H-5, H-31, H-1Xyl1
4	39.6 C			
5	52.6 CH	0.78 brd (12.0)	C: 10, 19, 30	H-1, H-3, H-7, H-31
6	20.9 CH_2_	1.61 m	-	
		1.41 m	-	H-30
7	27.7 CH_2_	1.60 m	-	H-15
		1.17 m	-	H-32
8	39.4 CH	3.15 brd (12.5)	-	H-15, H-19
9	150.6 C	-	-	-
10	39.2 C	-	-	-
11	110.9 CH	5.16 m	C: 10, 13	H-1
12	33.8 CH_2_	2.52 brd (16.8)	C: 18	H-17, H-32
		2.41 brdd (5.8; 17.0)	C: 11, 14, 18	H-21
13	59.0 C	-	-	-
14	42.9 C	-	-	-
15	43.7 CH_2_	2.21 dd (6.7; 11.5)	C: 13, 16, 17, 32	H-7, H-32
		1.35 brt (11.5)	C: 14, 16, 32	H-8
16	75.3 CH	5.72 q (9.6)	C: 20, OAc	H-32
17	52.5 CH	2.69 d (9.6)	C: 12, 13, 15, 18, 21	H-12, H-21, H-32
18	177.5 C	-	-	-
19	22.0 CH_3_	1.25 s	C: 1, 5, 9, 10	H-1, H-2, H-8, H-9, H-30
20	85.5 C	-	-	-
21	28.3 CH_3_	1.45 s	C: 17, 20, 22	H-12, H-17, H-22
22	37.9 CH_2_	2.08 m	-	-
		1.91 m	C: 20	-
23	22.8 CH_2_	1.47 m	-	-
		1.39 m	-	-
24	38.1 CH_2_	1.91 m	C: 22, 23, 25, 26, 27	H-26
25	145.4 C	-	-	-
26	110.7 CH_2_	4.73 brs	C: 24, 27	H-27
		4.72 brs	C: 24, 27	H-24
27	22.0 CH_3_	1.65 s	C: 24, 25, 26	H-26
30	16.4 CH_3_	0.95 s	C: 3, 4, 5, 31	H-2, H-6, H-19, H-31
31	27.9 CH_3_	1.14 s	C: 3, 4, 5, 30	H-3, H-5, H-30, H-1 Xyl1
32	21.0 CH_3_	0.88 s	C: 8, 13, 14, 15	H-7, H-12, H-15, H-16, H-17
OCOCH_3_	171.1 C	-	-	-
OCOCH_3_	21.3 CH_3_	2.10 s	C: OAc	-

**Table 10 marinedrugs-18-00394-t010:** ^13^C and ^1^H NMR chemical shifts and HMBC and ROESY correlations of the aglycone moiety of quadrangularisoside C_1_ (**7**). ^a^ Recorded at 176.03 MHz in C_5_D_5_N/D_2_O (4/1). ^b^ Recorded at 700.00 MHz in C_5_D_5_N/D_2_O (4/1).

Position	δ_C_ mult. ^a^	δ_H_ mult. (*J* in Hz) ^b^	HMBC	ROESY
1	35.7 CH_2_	1.34 m	-	H-11
		1.30 m	-	H-11
2	26.7 CH_2_	1.98 m	-	-
		1.80 brdd (2.7; 11.9)	-	H-19, H-30
3	88.8 CH	3.19 dd (3.6; 11.9)	C: 4, 30, 1 Xyl1	H-1, H-5, H-31, H-1Xyl1
4	39.3 C	-	-	-
5	47.7 CH	0.93 dd (4.6; 11.0)	C: 4, 10, 19, 30, 31	H-1, H-3, H-31
6	22.9 CH_2_	1.94 m	-	H-19, H-30, H-31
7	120.0 CH	5.61 m	-	H-15, H-32
8	145.5 C	-	-	-
9	46.9 CH	3.33 brd (14.6)	-	H-12, H-19
10	35.2 C	-	-	-
11	22.3 CH_2_	1.73 m	-	H-1
		1.48 m	-	H-32
12	31.1 CH_2_	2.10 m	C: 13, 14	H-17, H-32
13	59.1 C	-	-	-
14	47.1 C	-	-	-
15	43.3 CH_2_	2.55 dd (7.3; 11.9)	C: 13, 14, 17, 32	H-7, H-32
		1.62 m	C: 14, 16, 32	-
16	74.7 CH	5.85 q (9.1)	C: 13, OAc	H-32
17	54.3 CH	2.66 d (9.1)	C: 12, 13, 18, 21	H-12, H-21, H-32
18	179.8 C	-	-	-
19	23.6 CH_3_	1.11 s	C: 1, 5, 9, 10	H-1, H-2, H-6, H-9, H-30
20	85.5 C	-	-	-
21	27.8 CH_3_	1.53 s	C: 17, 20, 22	H-12, H-17, H-23, H-24
22	39.2 CH_2_	1.10 m	-	-
		0.99 m	-	-
23	22.4 CH_2_	1.27 m	-	-
		1.17 m	-	-
24	38.8 CH_2_	2.27 td (4.6; 13.7)	C: 25	-
		1.75 m	C: 20	-
25	28.0 CH	1.55 m	-	-
26	22.5 CH_3_	0.80 s	C: 24, 25, 27	-
27	22.0 CH_3_	0.79 s	C: 24, 25, 26	-
30	17.0 CH_3_	1.01 s	C: 3, 4, 5, 31	H-2, H-6, H-19, H-31
31	28.4 CH_3_	1.18 s	C: 3, 4, 5, 30	H-3, H-5, H-6, H-30, H-1 Xyl1
32	31.9 CH_3_	1.15 s	C: 8, 13, 14, 15	H-7, H-11, H-12, H-15, H-17
OCOCH_3_	170.1 C	-	-	-
OCOCH_3_	21.0 CH_3_	2.01 s	C: OAc	-

**Table 11 marinedrugs-18-00394-t011:** ^13^C and ^1^H NMR chemical shifts and HMBC and ROESY correlations of carbohydrate moiety of quadrangularisosides D (**8**), D_1_ (**9**), D_2_ (**10**), D_3_ (**11**), and D_4_ (**12**). *^a^* Recorded at 176.03 MHz in C_5_D_5_N/D_2_O (4/1). *^b^* Bold = interglycosidic positions. *^c^* Italic = sulfate position. *^d^* Recorded at 700.00 MHz in C_5_D_5_N/D_2_O (4/1). Multiplicity by 1D TOCSY.

Atom	δ_C_ mult. *^a,b,c^*	δ_H_ mult. *^d^* (*J* in Hz)	HMBC	ROESY
Xyl1 (1→C-3)				
1	104.6 CH	4.63 d (7.1)	C-3	H-3; H-3, 5 Xyl1
2	**81.9** CH	3.96 t (8.6)	C: 1 Qui2; 1, 3 Xyl1	H-1 Qui2; H-4 Xyl1
3	75.3 CH	4.23 t (8.6)	C: 2, 4 Xyl1	H-1, 5 Xyl1
4	*76.0* CH	4.97 m	-	H-2 Xyl1
5	64.1 CH_2_	4.79 dd (5.7; 12.1)	C: 1, 3, 4 Xyl1	-
		3.74 brt (10.7)	C: 1 Xyl1	H-1, 3 Xyl1
Qui2 (1→2Xyl1)				
1	103.5 CH	5.11 d (7.8)	C: 2 Xyl1	H-2 Xyl1; H-3, 5 Qui2
2	75.7 CH	4.00 t (8.7)	C: 1, 3 Qui2	H-4 Qui2
3	*81.5* CH	4.99 t (8.7)	C: 2, 4 Qui2	H-5 Qui2
4	**79.3** CH	3.82 t (8.7)	C: 1 Xyl3; 3, 5, 6 Qui2	H-1 Xyl3; H-2 Qui2
5	71.8 CH	3.65 dd (6.1; 9.6)	C: 1 Qui2	H-1, 3 Qui2
6	18.3 CH_3_	1.54 d (6.1)	C: 4, 5 Qui2	H-4, 5 Qui2
Xyl3 (1→4Qui2)				
1	102.8 CH	5.09 d (7.1)	C: 4 Qui2; 5 Xyl3	H-4 Qui2; H-3,5 Xyl3
2	73.3 CH	4.08 t (8.9)	C: 1, 3 Xyl3	-
3	**86.1** CH	4.13 t (8.9)	C: 1 MeGlc4; 2 Xyl3	H-1, 5 Xyl3
4	69.4 CH	4.01 m	C: 3 Xyl3	-
5	65.9 CH_2_	4.46 dd (5.3; 12.5)	C: 1, 3, 4 Xyl3	-
		3.78 dd (8.0; 12.5)	C: 1, 3, 4 Xyl3	H-1 Xyl3
MeGlc4 (1→3Xyl3)				
1	104.2 CH	5.12 d (8.0)	C: 3 Xyl3	H-3 Xyl3; H-3, 5 MeGlc4
2	74.0 CH	3.87 dd (7.1; 9.8)	C: 1, 3 MeGlc4	H-4 MeGlc4
3	85.2 CH	3.71 t (8.9)	C: 2, 4 MeGlc4, OMe	H-1 MeGlc4; OMe
4	*76.2* CH	4.85 t (8.9)	C: 3, 5, 6 MeGlc4	H-2, 6 MeGlc4
5	76.3 CH	3.83 m	-	-
6	61.7 CH_2_	4.48 dd (2.7; 13.3)	C: 5 MeGlc4	-
		4.29 dd (6.2; 11.8)	C: 5 MeGlc4	-
OMe	60.6 CH_3_	3.92 s	C: 3 MeGlc4	-

**Table 12 marinedrugs-18-00394-t012:** ^13^C and ^1^H NMR chemical shifts and HMBC and ROESY correlations of carbohydrate moiety of quadrangularisoside E (**13**). *^a^* Recorded at 176.03 MHz in C_5_D_5_N/D_2_O (4/1). *^b^* Bold = interglycosidic positions. *^c^* Italic = sulfate position. *^d^* Recorded at 700.00 MHz in C_5_D_5_N/D_2_O (4/1). Multiplicity by 1D TOCSY.

Atom	δ_C_ mult. *^a,b,c^*	δ_H_ mult. *^d^* (*J* in Hz)	HMBC	ROESY
Xyl1 (1→C-3)				
1	104.7 CH	4.67 d (7.3)	C: 3; C: 5 Xyl1	H-3; H-3, 5 Xyl1
2	**82.2** CH	3.94 t (8.2)	C: 1 Qui2; 3 Xyl1	H-1 Qui2
3	74.6 CH	4.23 t (9.1)	C: 2, 4 Xyl1	H-1, 5 Xyl1
4	*75.6* CH	4.96 m	C: 3 Xyl1	H-2 Xyl1
5	63.7CH_2_	4.75 dd (5.5; 11.0)	C: 1, 3, 4 Xyl1	-
		3.73 t (10.1)	-	H-1 Xyl1
Qui2 (1→2Xyl1)				
1	104.5 CH	4.92 d (7.6)	C: 2 Xyl1; 5 Qui2	H-2 Xyl1; H-3, 5 Qui2
2	75.3 CH	3.83 t (9.1)	C: 1, 3 Qui2	-
3	74.9 CH	3.89 t (9.1)	C: 2, 4 Qui2	H-1, 5 Qui2
4	**87.1** CH	3.41 t (9.1)	C: 1 Glc3; 3, 5 Qui2	H-1 Glc3; H-6 Qui2
5	71.3 CH	3.60 m	C: 1, 3, 4 Qui2	H-1 Qui2
6	17.7 CH_3_	1.59 d (6.0)	C: 4, 5 Qui2	H-4, 5 Qui2
Glc3 (1→4Qui2)				
1	104.2 CH	4.76 d (7.3)	C: 4 Qui2	H-4 Qui2; H-3,5 Glc3
2	73.5 CH	3.84 t (8.9)	C: 1, 3 Glc3	H-4 Glc3
3	**86.0** CH	4.17 t (8.9)	C: 1 MeGlc4; 2, 4 Glc3	H-1 MeGlc4, H-1, 5 Glc3
4	69.2 CH	3.76 t (8.9)	C: 3, 5, 6 Glc3	H-6 Glc3
5	74.8 CH	4.11 t (8.9)	-	H-1, 3 Glc3
6	*67.5* CH_2_	4.99 d (9.7)	C: 4 Glc3	-
		4.56 dd (6.3; 11.3)	C: 5 Glc3	-
MeGlc4 (1→3Glc3)				
1	104.4 CH	5.19 d (7.7)	C: 3 Glc3, 5 MeGlc4	H-3 Glc3; H-3, 5 MeGlc4
2	74.0 CH	3.87 t (7.7)	C: 1, 3 MeGlc4	H-4 MeGlc4
3	85.2 CH	3.72 t (9.6)	C: 2, 4 MeGlc4, OMe	H-1 MeGlc4; OMe
4	*76.1* CH	4.87 t (9.6)	C: 3, 5, 6 MeGlc4	H-2 MeGlc4
5	76.4 CH	3.85 t (9.6)	-	H-1 MeGlc4
6	61.6 CH_2_	4.48 d (11.5)	C: 5 MeGlc4	-
		4.31 dd (5.8; 11.5)		
OMe	60.6 CH_3_	3.92 s	C: 3 MeGlc4	

**Table 13 marinedrugs-18-00394-t013:** The cytotoxic activities of glycosides **1**–**13** and cladoloside C (positive control) against mouse erythrocytes, neuroblastoma Neuro 2a cells, and normal epithelial JB-6 cells. The inhibiting concentration of the glycosides **1**–**13** on cell viability (IC_50_) and colony formation (ICCF_50_) of HT-29 cells.

Glycoside	ED_50_, µM	Cytotoxicity EC_50_, µM		IC_50_, µM	ICCF_50_, µM
	Erythrocytes	Neuro-2a	JB-6	HT-29	HT-29
quadrangularisoside A (**1**)	1.57 ± 0.16	27.43 ± 2.23	>50.00	>20	18.3 ± 0.8
quadrangularisoside A_1_ (**2**)	1.11 ± 0.08	21.34 ± 1.32	27.91 ± 2.19	>20	7.2 ± 0.3
quadrangularisoside B (**3**)	0.51 ± 0.05	9.79 ± 0.76	18.62 ± 0.80	0.49 ± 0.03	0.28 ± 0.04
quadrangularisoside B_1_ (**4**)	0.23 ± 0.02	11.52 ± 0.26	17.01 ± 0.81	0.48 ± 0.01	0.25 ± 0.07
quadrangularisoside B_2_ (**5**)	0.51 ± 0.01	16.27 ± 1.70	19.43 ± 0.88	4.8 ± 0.1	0.46 ± 0.05
quadrangularisoside C (**6**)	0.56 ± 0.02	22.34 ± 0.82	11.35 ± 0.85	12.6 ± 1.5	0.65 ± 0.08
quadrangularisoside C_1_ (**7**)	0.11 ± 0.01	14.68 ± 0.33	9.13 ± 0.37	19.0 ± 2.0	1.38 ± 0.4
quadrangularisoside D (**8**)	0.24 ± 0.01	10.98 ± 0.78	12.71 ± 1.76	3.0 ± 0.2	0.84 ± 0.07
quadrangularisoside D_1_ (**9**)	0.41 ± 0.02	10.68 ± 0.94	16.50 ± 2.32	4.1 ± 0.2	1.31 ± 0.08
quadrangularisoside D_2_ (**10**)	3.31 ± 0.09	24.49 ± 1.09	37.87 ± 1.51	>20	6.75 ± 1.25
quadrangularisoside D_3_ (**11**)	6.03 ± 0.42	>50.00	>50.00	>20	>20
quadrangularisoside D_4_ (**12**)	5.45 ± 0.15	>50.00	>50.00	>20	>20
quadrangularisoside E (**13**)	2.04 ± 0.05	19.48 ± 1.47	19.01 ± 0.47	16.8 ± 0.4	1.87 ± 0.94
cladoloside C	0.20 ± 0.02	16.24 ± 2.41	11.20 ± 0.27	---	---

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
