# Peer review of "Structures and Bioactivities of Quadrangularisosides A, A1, B, B1, B2, C, C1, D, D1–D4, and E from the Sea Cucumber Colochirus quadrangularis: The First Discovery of the Glycosides, Sulfated by C-4 of the Terminal 3-O-Methylglucose Residue. Synergetic Effect on Colony Formation of Tumor HT-29 Cells of these Glycosides with Radioactive Irradiation"

_marinedrugs, 2020, doi:10.3390/md18080394_

Round 1
Reviewer 1 Report
The manuscript contains some interesting and current research on the isolation and structure elucidation of quadrangularisosides, as well as on bioactivity of these compounds. As regards the scientific content and the formal arrangement of the manuscript, they are both satisfactory and I can recommend to accept the manuscript for publication.
Author Response
We are very grateful to the Reviewer for positive review of our manuscript.
Reviewer 2 Report
In the manuscript entitled "Structures and bioactivities of quadrangularisosides A, A1, B, B1, B2, C, C1, D, D1–D4, and E from the sea cucumber Colochirus quadrangularis. The first discovery of the glycosides, sulfated by C-4 of the terminal 3-O-methylglucose residue. Synergetic effect on colony formation of tumor HT-29 cells of these glycosides with radioactive irradiation", by Silchenko et al. (Manuscript marinedrugs-870840), the authors describe the discovery of
13 novel mono-, di- and trisulfated triterpene glycosides, quadrangularisosides A isolated from the sea cucumber Colochirus quadrangularis. Some of the characterized compounds have a new aglycones having 25-hydroxperoxyl or 24-hydroxperosxyl groups. By their characterization of these novel compounds by 2D NMR spectroscopy and HR-ESI, the authors not only provided a very detailed and clear elucidation of the novel structure, but also are able to clarify and corrected earlier published structures from other groups.
Moreover the authors demonstrate that all compounds are rather strong hemolytics.
To the reviewer’s point of view the current version of the manuscript is very well written and comprehensive.
Other more specific comments/questions:
- line 50 to76: I would like the authors to suggest to be a little bit more diplomatic concerning earlier studies on the structures/interpretation of data. Maybe the wrongly assigned NMR signal etc. is rather something for the discussion, but not for the second paragraph of the introduction.
- Maybe the authors could briefly describe in the introduction, the function of the hydroxperosxyl functions of their molecules. What is chemically happening upon exposure with radiation? What is a possible mode of function in cancer therapy?
- Table 13: Can the authors provide errors for their measurements? How many replicates?
Author Response
We are very grateful to the Reviewer for the comments and questions.
- We agree with the Reviewer,s suggestion to be a little bit more diplomatic concerning earlier studies on the structures/interpretation of data. The discussion of clarification and correction of the earlier published structures was replaced from the Introduction of the manuscript to the first paragraph of Results and Discussion. All corrections are highligted through the text.
- "Maybe the authors could briefly describe in the introduction, the function of the hydroperoxyl functions of their molecules." We suppose the glycosides having hydroperoxyl group in the side chains of their aglycones are the intermediates or by-products in the biosynthetic transformations of the side chains. This supposition is justified by the absence of membranolytic activity of these glycosides and their minor amounts in the organism-producer, so they are obviously not the target biosynthetic products. Therefore we think there is no necessity to discuss the function of such glycosides in the manuscript.
- "What is chemically happening upon exposure with radiation?" When the Soft agar assay is used the glycosides are accumulated inside the cells, so they are protected by the cell membranes from the direct action of radiation. Hence they are not the target molecules that are chemically transformed upon radiation exposure.
- "What is a possible mode of function in cancer therapy?"
- "Table 13: Can the authors provide errors for their measurements? How many replicates?" All these data are added to the text of the manuscript.
Reviewer 3 Report
Manuscript by Silchenko et al. entitled “Structures and bioactivities of quadrangularisosides A, A1, B, B1, B2, C, C1, D, D1–D4, and E from the sea cucumber Colochirus quadrangularis. The first discovery of the glycosides, sulfated by C-4 of the terminal 3-O-methylglucose residue. Synergetic effect on…” is dedicated to structure-activity relationship of multiple triterpene glycosides isolated from tropical sea cucumbers. The authors performed excellent isolation and structure elucidation work that can be useful from taxonomical point of view. The authors also determined structure-activity relationship between identified molecules using hemolytic, cytotoxic, and soft agar in vitro models.
In overall, this is a very good work that needs just a few final improvements before publishing.
Major points.
Lines 579-580. There is one methodological aspect that should be revisited by the authors that impacts author’s conclusion point. When comparing compound’s activities in different settings, it is important to consider that most molecules do not behave as ions (that are subject of tight control by different ion channels and pumps) but accumulate inside the cell. Therefore, one has to think about the amount of compound per cell that was given during experiment and not just about provided concentrations. Concentration of molecule may be the same but its total amount will be different in 0.1 ml versus 1.0 ml. This notion specifically relates to experiments that evaluate compound’s effects on colony formation and relate this data to previously determined IC50 values. According to my calculations, the total volume used for soft agar experiment and cytotoxicity experiments as well as number of cells seeded per well are different, and the total amount of compounds per cell in soft agar experiments is about 10-times higher in comparison to cytotoxicity experiments. Therefore, the observed lower values needed to inhibit colony formation can be simply due to higher amount of compound per cell (hence, increased cytotoxicity) and not due to hypothetical signaling properties of compounds. Please, reconsider your evaluation OR provide direct evidence for signaling effects of studied compounds. This also determines what you write in the Abstract lines 38-39.
Line 559, Table 13. You must express your data as Mean +/- SEM (SD) and indicate the number of provided experiments. In the current form, this table is unpublishable.
Line 595, Figure 2. Please, provide significance and p values for the difference for each compound when the difference from Irradiation is significant.
Minor points.
Lines 40, 590. Change 0,02 to 0.02
Line 528. Replace “As regards the…” by “As to the…”
Line 542. Replace “analogical” to “similar”
Lines 785-792. Volume of the assay needs to be indicated.
Lines 802-803. This sentence is unclear/grammatically incorrect.
Lines 822-844. This sentence need correction as follows: “Three hours later, cells were harvested and used for soft agar assay to establish the synergism between irradiation and investigated compounds.”
Lines 819-824. Volume of the assay needs to be indicated.
Lines 869-870. The beginning of the sentence should be improved, something like “ It is known that sea cucumber triterpene glycosides are taxonomically specific and can be used as…”
Author Response
We are very appreciated to the Reviewer for the revision of the manuscript. All minor points, indicated by Reviewer are fixed and highlighted in the text.
As to the major points:
- We added the information to the Materials and Methods concerning the volumes used for each experiment: cytotoxicity (1.0×104/200µL, Line 781), soft agar assay (2.4×104/mL, Line 803) and cell irradiation (5.0×105/5 mL, Line 814). These amounts of cells (from 5×104/mL=1.0×104/200µL to 2.4x104/mL) were chosen based on peculiarities of different assays protocols. We can’t use low amounts of cells for soft agar assay because it is spontaneous process for cancer cells and not all the cells will grow in colonies in soft agar. Moreover, such significant differences of the compound’s activities in different settings (cytotoxicity and soft agar assays), for example: (6), IC50 = 12.6 mM and ICCF50 = 0.65 mM; (3), IC50 = 0.49 mM and ICCF50 = 0.28 mM cannot be fully explained only by different amounts of glycosides molecules and, in our opinion, are concerned with SARs and may be different mechanisms of action.
- The data in Table 13 are corrected and expressed as Mean +/- SD, all the experiments were made in triplicate, that is depicted in the text of the manuscript.
- p-values are provided in Figure 2 and in its legend and to the corresponding paragraphs of Materials and Methods.